# RNA targeting unleashes indiscriminate nuclease activity of CRISPR–Cas12a2

Jack P. K. Bravo[1,8], Thomson Hallmark[2,8], Bronson Naegle[2], Chase L. Beisel[3,4], Ryan N. Jackson[2✉] & David W. Taylor[1,5,6,7✉]

Cas12a2 is a CRISPR-associated nuclease that performs RNA-guided, sequence-nonspecific degradation of single-stranded RNA, single-stranded DNA and double-stranded DNA following recognition of a complementary RNA target, culminating in abortive infection[1]. Here we report structures of Cas12a2 in binary, ternary and quaternary complexes to reveal a complete activation pathway. Our structures reveal that Cas12a2 is autoinhibited until binding a cognate RNA target, which exposes the RuvC active site within a large, positively charged cleft. Double-stranded DNA substrates are captured through duplex distortion and local melting, stabilized by pairs of 'aromatic clamp' residues that are crucial for double-stranded DNA degradation and in vivo immune system function. Our work provides a structural basis for this mechanism of abortive infection to achieve population-level immunity, which can be leveraged to create rational mutants that degrade a spectrum of collateral substrates.

Prokaryotic adaptive immunity typically uses CRISPR–Cas systems to target and degrade foreign genetic elements, including phage and plasmids[2,3]. However, it was recently discovered that Cas12a2 from *Sulfuricurvum* sp. PC08-66 instead relies on abortive infection—that is, dormancy or cell death in response to the presence of an invader—to achieve population-level immunity, preventing the replication and transmission of plasmids[1].

Although Cas12a2 sometimes co-occurs with Cas12a systems in bacteria and can use Cas12a CRISPR RNA (crRNA), Cas12a2 recognizes an RNA target strand with a suitable protospacer-flanking sequence (PFS; for example, 5′-GAAAG-3′) rather than the double-stranded (ds)DNA target of Cas12a[1,4,5]. Furthermore, Cas12a2 is immune to the effects of many anti-CRISPR proteins that target Cas12a, and aside from a conserved RuvC nuclease domain and pre-crRNA processing region, Cas12a and Cas12a2 sequences bear little resemblance to one another (about 10–20% sequence identity). Notably, Cas12a2 lacks a Nuc domain (involved in DNA target strand loading), but instead contains a zinc ribbon (ZR), and it contains a unique insertion domain in place of the Cas12a bridge helix.

Unlike many of the recently characterized abortive infection systems[6–14], Cas12a2 does not rely on the production of secondary messengers to achieve antiphage immunity. Instead, Cas12a2 activation induces robust, nonspecific cleavage of single-stranded (ss)RNA, ssDNA and dsDNA[1] in *trans*. This mechanism is unique to Cas12a2, although the molecular basis for collateral nucleic acid degradation is unknown.

To understand the unique mechanisms of activation, substrate capture and indiscriminate nuclease activity underlying the function of Cas12a2, we carried out biochemical, structural and in vivo analyses, including determining cryo-electron microscopy (cryo-EM) structures of autoinhibited Cas12a2–crRNA (binary complex) associated with an RNA target (ternary complex) and bound to both an RNA target and a dsDNA collateral substrate mimetic (quaternary complex).

## Structure of Cas12a2 binary complex

To gain insights into the function of Cas12a2, we first purified a binary complex consisting of catalytically active Cas12a2 and a mature crRNA and determined the structure using cryo-EM to a global resolution of 3.2 Å (Extended Data Table 1). The quality of the map allowed de novo modelling of most of Cas12a2, aside from the flexible PFS-interacting (PI) and ZR domains.

Cas12a2 adopts a bilobed architecture (Fig. 1), with a recognition (REC) lobe comprising the REC1 and REC2 domains and the nuclease (NUC) lobe consisting of the PI, wedge (WED), RuvC nuclease, ZR and Cas12a2-specific insertion domains. The overall architecture of the binary complex resembles an oyster, as opposed to the triangular 'sea conch' shape of Cas12a[15].

Although Cas12a and Cas12a2 share low (10–20%) sequence similarity, comparison of their WED and RuvC domains shows a high degree of structural similarity (root mean square deviation of 1.073 Å across 120 equivalent residues with FnCas12a; PDB ID 5NG6; Extended Data Fig. 2). Furthermore, the crRNA 5′ stem–loop is in an identical configuration in both complexes, and a loop containing basic residues is similarly positioned to catalyse pre-crRNA maturation. The common 'chassis' formed by these domains provides a structural scaffold that enables the same crRNAs to prime and guide either complex to the same target sequences for their different functions.

Despite the similar domain organization to Cas12a within the WED and RuvC domains, Cas12a2 has a unique α-helical REC lobe, with no known structural homologues. The differences in the structural organization of the REC lobe probably allow Cas12a2 to escape targeting by many anti-CRISPR proteins that can efficiently shut down Cas12a[1] (Extended Data Fig. 3). Of note, seven nucleotides of pre-ordered crRNA sit at the interface between REC1 and REC2 in a conformation in which

[1]Department of Molecular Biosciences, University of Texas at Austin, Austin, TX, USA. [2]Department of Chemistry and Biochemistry, Utah State University, Logan, UT, USA. [3]Helmholtz Institute for RNA-based Infection Research (HIRI), Helmholtz-Centre for Infection Research (HZI), Würzburg, Germany. [4]Medical Faculty, University of Würzburg, Würzburg, Germany. [5]Interdisciplinary Life Sciences Graduate Program, University of Texas at Austin, Austin, TX, USA. [6]Center for Systems and Synthetic Biology, University of Texas at Austin, Austin, TX, USA. [7]LIVESTRONG Cancer Institutes, Dell Medical School, Austin, TX, USA. [8]These authors contributed equally: Jack P. K. Bravo, Thomson Hallmark. ✉e-mail: ryan.jackson@usu.edu; dtaylor@utexas.edu

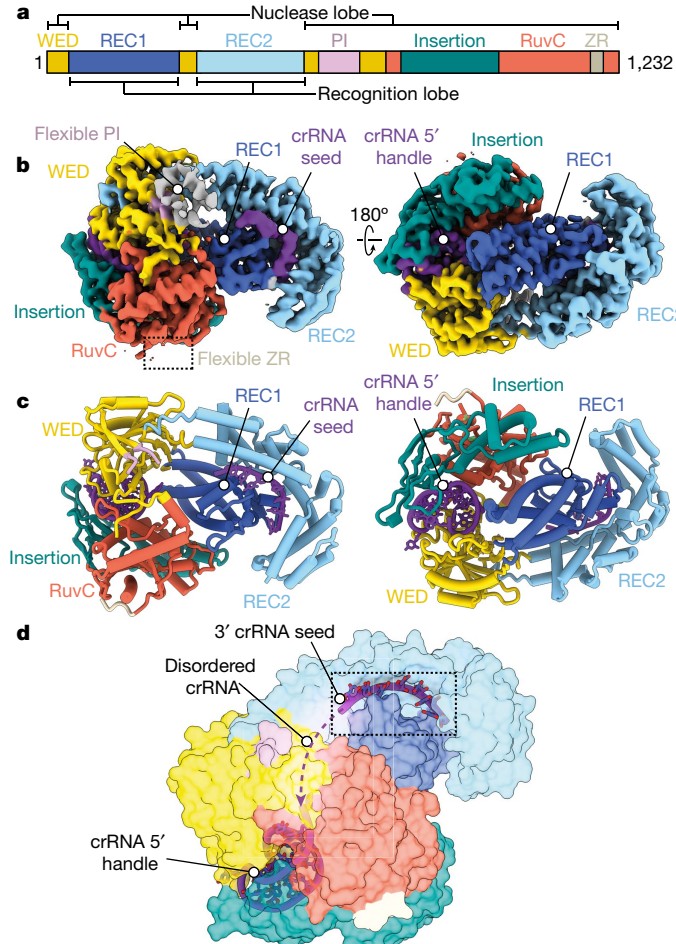

**Fig. 1 | Cas12a2 binary complex resembles an oyster and orders the crRNA.**
**a**, Domain organization of Cas12a2. **b**, Cryo-EM structure of Cas12a2 binary
complex coloured by structural domain as in **a. c**, Atomic model of Cas12a2
binary complex. **d**, Putative 3′ seed region of crRNA. Seven bases from the 3′
end are ordered, and bases are solvent-exposed, probably acting as a seed for
target RNA binding.

bases are solvent-exposed and primed for target recognition (Fig. 1d).
This region is towards the 3′ end of the crRNA, and the intervening
sequence between the 5′ crRNA stem–loop and this pre-ordered guide
is disordered in our structure, probably owing to flexibility. This is in
contrast to the case for Cas12a, which has a well-described pre-ordered
crRNA immediately flanking the 5′ stem–loop. In Cas12a, this region is
highly sensitive to mismatches because it initiates R-loop formation
following recognition of the protospacer adjacent motif (PAM) and
is considered a seed region[16]. By contrast, Cas12a2 is insensitive to
single mismatches within the entirety of the crRNA but has reduced
in vivo activity when truncated on the 3′ end[1]. Our structure suggests
that Cas12a2–crRNA–target strand (TS) duplex formation may initiate
and propagate from the 3′ end of the crRNA, enabling Cas12a2 to target
phage that have escaped surveillance by Cas12a through mutagenesis
of the PAM or 5′ seed regions.

## RNA target binding activates Cas12a2

To understand how RNA target recognition is distinct from other Cas12
family nucleases, we next investigated how RNA target binding acti-
vates Cas12a2. We determined a 2.9-Å-resolution cryo-EM structure
of a ternary complex consisting of Cas12a2, crRNA and a target ssRNA
containing the non-self PFS 5′-GAAAG-3′ (Fig. 2).

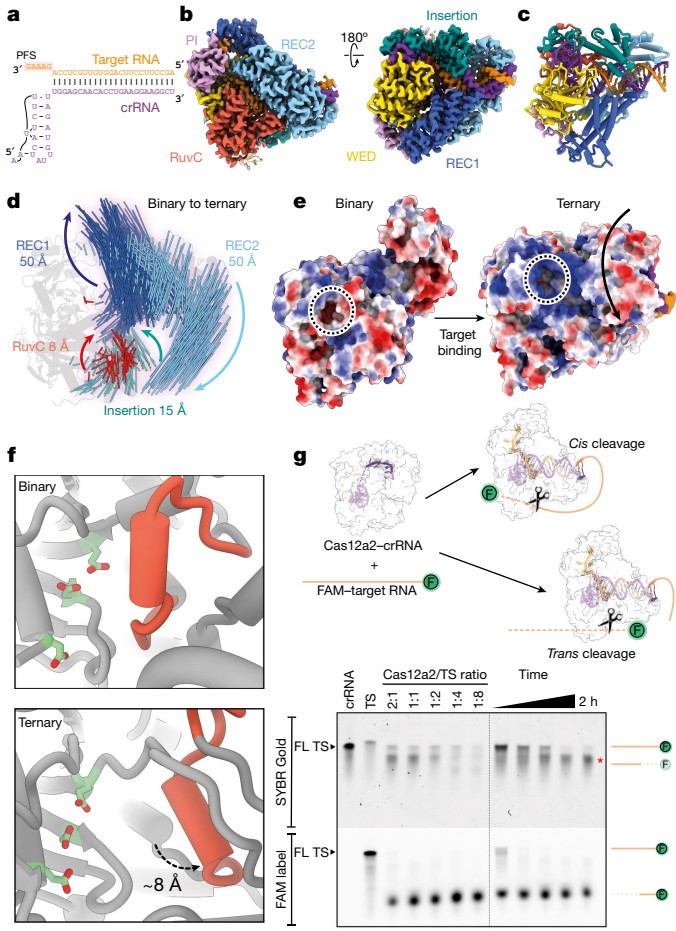

**Fig. 2 | Target binding leads to large-scale structural arrangements for
activation. a**, Schematic of crRNA–TS duplex. PFS on target RNA is highlighted.
**b**, Cryo-EM structure of Cas12a2 ternary complex. **c**, Atomic model of Cas12a2
ternary complex. **d**, Motion vector map showing conformational changes of
Cas12a2 induced following ternary complex formation. Binary complex model
shown as grey cartoon. Conformational changes induced by target RNA binding
are also shown in Supplementary Video 1. **e**, Surface electrostatic potential of
binary and ternary complex, showing how active site (dotted circle) becomes
exposed following ternary complex formation. This is accompanied by the
formation of a large positively charged groove adjacent to the activate site.
**f**, Displacement of RuvC gating helix (red) by about 8 Å following ternary complex
formation exposes active site residues (light green). **g**, Target protection
revealed by SYBR staining, showing only peripheral cleavage of target RNA
when binary complex is in molar excess, and total degradation when target is
in excess. The observed protection persisted for 2 h with excess binary complex.
Representative of three independent experiments with similar results. For gel
source data, see Supplementary Fig. 1. FL, full length.

The 22-base-pair A-form crRNA–target RNA duplex runs through
the centre of the complex. At the 5′ end of the crRNA guide, the duplex
splits with the crRNA 5′ stem–loop wedged between the RuvC and WED
domains and the 3′ PFS end of the target RNA is gripped by the PI domain,
which has now become ordered (Fig. 2b). Each of the five nucleotides of
the PFS make specific base contacts with residues within the PI domain,
including hydrogen bonding and π–π stacking (Extended Data Fig. 4).
These contacts stabilize the otherwise flexible PI domain, allowing
Cas12a2 to distinguish self (that is, complementary to the crRNA 5′ han-
dle) from non-self target RNA on the basis of the PFS. Removal of the PI
domain had no effect on the overall structure of Cas12a2, but prevented
activation of nuclease activity (Extended Data Fig. 4). Notably, this is a
completely distinct mechanism of self-versus-non-self discrimination

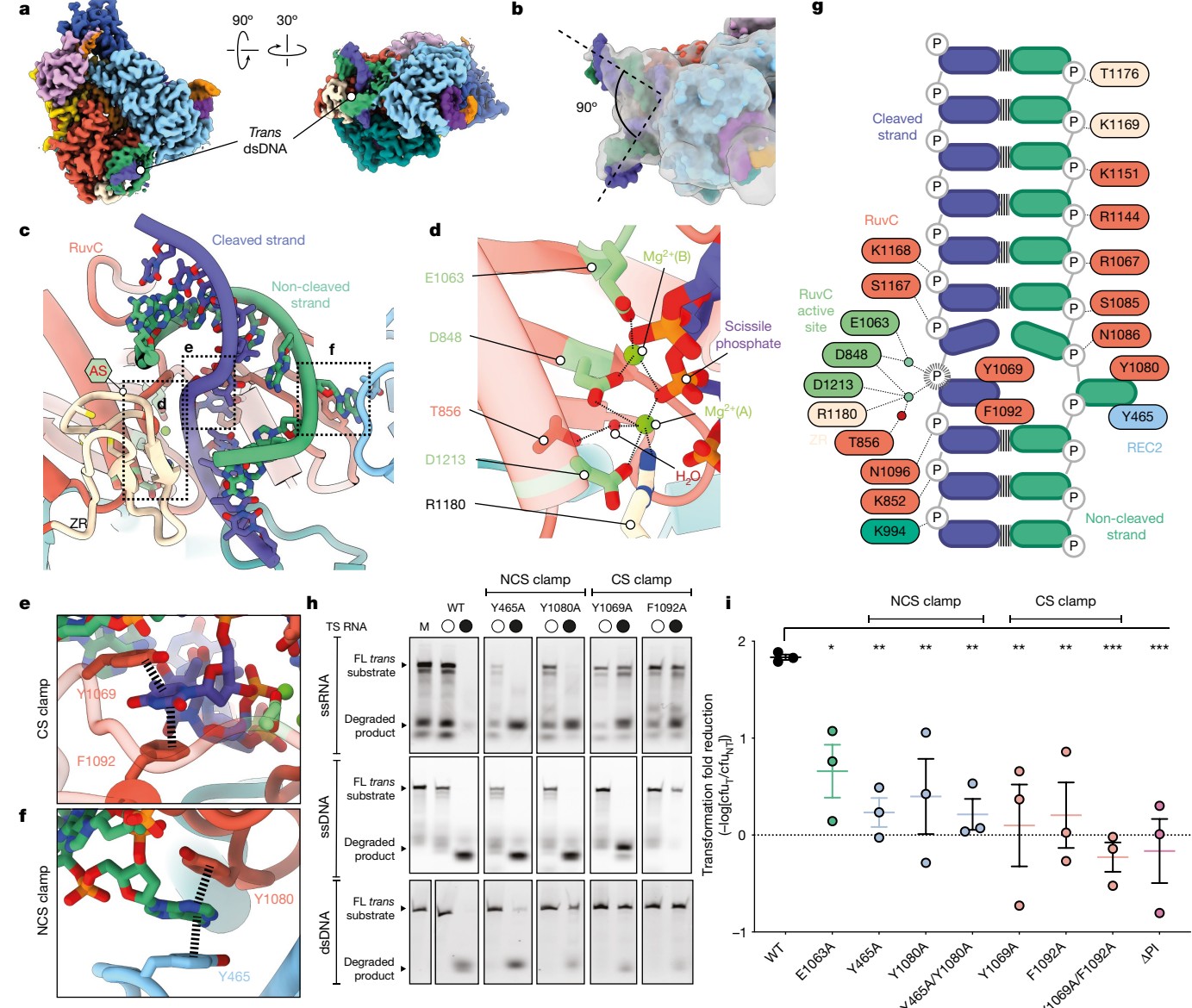

**Fig. 3 | Cas12a2 binds and clamps duplex DNA. a**, Cryo-EM structure of Cas12a2 quaternary complex. *Trans*-dsDNA is shown as slate blue and sea green. **b**, Atomic model of Cas12a2 quaternary complex, with 5 Å low-pass filtered map shown in transparent grey. To highlight the 90° kink in the collateral duplex, two linear ideal B-form dsDNA models have been rigid-body fitted into the map. **c**, dsDNA situated within active site. **d**, Close-up view of Cas12a2 active site. **e**, CS held in place through aromatic clamp. **f**, NCS held in place through aromatic clamp. **g**, Schematic of interactions between Cas12a2 and the collateral dsDNA substrate. CS scissile phosphate is denoted by dashed outline. **h**, Cleavage of ssRNA (top), ssDNA (middle) and dsDNA (bottom) by Cas12a2,

and aromatic clamp mutants. M, size marker (intact substrate). Representative of three independent experiments with similar results. **i**, Alterations to essential residues resulted in loss of the ability of Cas12a2 to clear plasmid (that is, lower transformation fold reduction, calculated as $-\log_{10}[\text{cfu}_T/\text{cfu}_{NT}]$, in which $\text{cfu}_T$ and $\text{cfu}_{NT}$ represent the number of colony-forming units for target and non-target plasmids, respectively). Significance between WT and mutant SuCas12a2 was determined by two-sided Student *t*-test. *$P < 0.05$, **$P < 0.01$, ***$P < 0.001$. Experiments were carried out in triplicate, and error bars correspond to the mean and standard error. For gel source data, see Supplementary Fig. 1.

from that of Cas13 and several type III effector complexes, for which nuclease activity is inhibited by additional complementarity with the 5′ crRNA tag region of the crRNA[17–19]. By contrast, Cas12a2 is exclusively activated following recognition of an appropriate PFS sequence.

Cas12a2 is unable to degrade nucleic acids in the absence of a suitable target RNA[1] (Extended Data Fig. 4). Superposition of the binary and ternary complexes reveals substantial conformational changes localized to the REC1 and REC2 domains, whereas the NUC lobe remains predominantly static (Fig. 2d). REC1 and REC2 are both displaced by up to about 50 Å and move in different directions, creating a central channel, which accommodates the crRNA–TS duplex. The insertion domain

moves by up to about 15 Å, but these changes are exclusively localized to the carboxy-terminal half of the domain (residues 938–1,030). The amino-terminal half (residues 870–937), which makes numerous contacts with the crRNA 5′ stem–loop, remains static. On the basis of this observation, we propose that the insertion domain acts as a transducer, allowing allosteric communication between the REC and NUC lobes in response to target RNA binding (Extended Data Fig. 5).

Inspection of the RuvC active site in the autoinhibited binary complex reveals that the catalytic triad (D848, E1063 and D1213) is buried within a solvent-excluded pocket (Fig. 2e). Strikingly, the conformational changes that accompany target RNA binding create

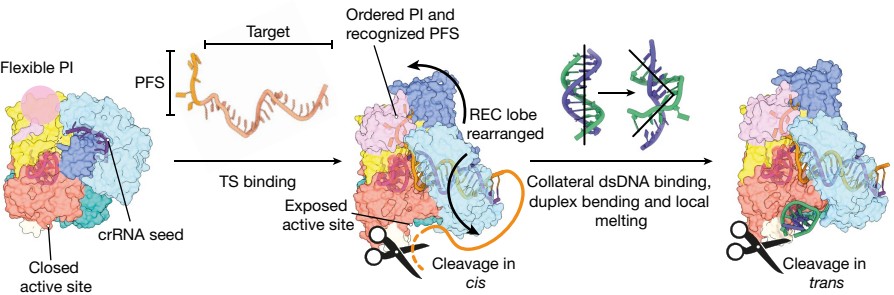

**Fig. 4 | Mechanism of RNA target-activated collateral nuclease activity in Cas12a2.** In the binary complex (Cas12a2–crRNA), the active site is occluded. The 3′ end of the crRNA is a pre-ordered seed, and the PI domain is flexible. Binding of a complementary RNA target strand (TS) containing a PFS triggers significant conformational changes within the REC lobe, exposing the RuvC active site to facilitate TS RNA trimming in *cis* and nuclease activity in *trans*. This enables binding of dsDNA in *trans*, resulting in duplex kinking and local melting. The duplex is held open through the action of two pairs of aromatic clamp residues (Fig. 3), enabling nicking by RuvC.

a 25-Å-wide positively charged groove that exposes the active site (Fig. 2e). This groove is of sufficient size to accommodate both single- and double-stranded nucleic acids. Although other Cas12 proteins undergo conformational changes following crRNA hybridization (up to about 25 Å for Cas12a[20] and Cas12j[21], but more typically up to about 10 Å (ref. [22])), the approximately 50-Å conformational rearrangements we observe for Cas12a2 are considerably larger, highlighting the distinct activation mechanism of Cas12a2 (Supplementary Video 1).

Access to the RuvC catalytic triad is also mediated by the approximately 8-Å shift of a lid helix, which contributes to the change in active site solvent exposure (Fig. 2f). This is akin to the lid loop or helix that gates active site exposure reported for other Cas12 endonucleases[23,24].

Once activated by target binding, the lid of these endonucleases remains 'open', enabling ssDNA cleavage in *trans*. However, in previously reported Cas12a structures (including FnCas12a2, AsCas12a and LbCas12a), the RuvC active site is buried owing to the presence of the Nuc domain[25–28]. In structures of catalytically dead Cas12b and Cas12i, bystander ssDNA bound in *trans* is tightly interwoven to sit within the active site[22,23] (Extended Data Fig. 6). This is in contrast to the highly accessible Cas12a2 RuvC active site in the ternary complex, providing a structural basis for efficient cleavage of a wide range of substrates in *trans*. The lack of a Nuc domain and the presence of a highly exposed RuvC active site in the ternary structure thus explain why Cas12a2 collateral nuclease activation results in an abortive infection phenotype[1] whereas Cas12a collateral ssDNase activity does not play a role in bacterial immunity[29]. The Nuc domain may act as a physical barrier to the RuvC active site in other Cas12 enzymes (including Cas12a), limiting cleavage in *trans*.

Unlike in Cas12a, the path followed by the target RNA–crRNA duplex completely circumvents the RuvC active site, suggesting a mechanism for RNA degradation in *trans*. To test this, we incubated the Cas12a2 binary complex with fluorescently labelled target RNA at a range of molar ratios and analysed RNA cleavage. The 5′ fluorescein label was consistently trimmed owing to the approximately 20-nucleotide flexible RNA sequence extending from the spacer (Fig. 2g). SYBR Gold staining revealed that even though the extended single-stranded 5′ end of the target RNA was trimmed, the target RNA otherwise remained intact and was protected from degradation with a molar equivalence or excess of Cas12a2. This is distinct from the cleavage mechanisms of other Cas12 nucleases that achieve antiphage immunity through cleavage in *cis*[16,29], and is reminiscent of Cas13 RNase activity in *cis* and *trans*, for which the hybridized region of the target RNA remains intact[18].

## Collateral dsDNA binding through duplex contortion

We next sought to visualize how Cas12a2 can accommodate and degrade nucleic acid duplexes. To this end, we determined a 2.7-Å-resolution structure of crRNA-guided Cas12a2 bound both to an activating target RNA and to collateral phosphorothioate dsDNA substrate (Fig. 3). The RuvC active site and 11 of 20 base pairs of the DNA duplex were well resolved, whereas the flexible DNA ends are visible only at lower density thresholds.

In our structure, the dsDNA duplex is sharply bent by about 90° (Fig. 3b), resulting in duplex distortion and local melting of two base pairs in the immediate vicinity of the RuvC active site (Fig. 3c), enabling the positioning of the scissile phosphate adjacent to the RuvC catalytic triad (Fig. 3d). We designate the DNA strand within the RuvC active site as the cleaved strand (CS) and its complement DNA strand as the non-cleaved strand (NCS) to differentiate from target and non-target strand nomenclature used to describe the strands of dsDNA that are specifically targeted with a crRNA guide (for example, Cas12a or Cas9). As both DNA strands contained non-hydrolysable phosphorothioate modifications, we could visualize the pre-hydrolysis RuvC active site state, including two $Mg^{2+}$ ions and a putative activating water adjacent to one of the ions (designated $Mg^{2+}(A)$; Fig. 3d and Extended Data Fig. 7).

Adjacent to the RuvC active site, both the CS and NCS are stabilized by a large network of nonspecific interactions with Cas12a2 (Fig. 3c,g). This hub of contacts with both duplex ends induces duplex bending and local melting. The melted bases are subsequently captured by two pairs of 'aromatic clamps' (Y465 and Y1080, Y1069 and F1092, respectively) that each hold a single DNA base through π–π stacking, preventing rehybridization (Fig. 3e,f). We confirmed this result through in vitro collateral nuclease assays (Fig. 3h): whereas wild-type (WT) Cas12a2 degraded ssRNA, ssDNA and dsDNA in *trans* when activated with complementary target RNA, the NCS clamp alterations Y465A and Y1080A seemed to reduce duplex cleavage, while having no effect on ssRNase or ssDNase nuclease activity. This indicates that unwinding by the NCS clamp is critical for nuclease activity of a DNA duplex. CS clamp alterations Y1069A and F1092A abrogate dsDNase activity as expected. Notably, whereas both CS clamp alterations prevent ssRNase activity, only F1092A blocks ssDNase activity (Fig. 3h). To better understand how these mutants would affect in vivo activity, we further tested these point mutations on Cas12a2 supercoiled plasmid DNA cleavage activity, and found that all had severely reduced activity (Extended Data Fig. 8). Of note, the NCS(Y465A) mutant that can cleave a linear FAM-labelled substrate (Fig. 3h) is also able to nick and/or linearize supercoiled plasmid, but is unable to degrade DNA in a timescale similar to that of the WT, providing a rationale for the in vivo effects of this mutant.

As Y1069A preserves ssDNase activity but prevents ssRNase and dsDNase activity, this mutant may enable development of a molecular biosensor that degrades a fluorescence reporter ssDNA following recognition of a complementary ssRNA, enabling sensitive detection of RNA without target depletion as is the case in Cas13 reporter systems.

As NCS clamp alterations ablate dsDNase activity but have no effect on single-stranded collateral nuclease activity, we tested

these separation-of-function mutants in vivo. As expected, CS alterations and PI domain truncation blocked Cas12a2 activity (Fig. 3i and Extended Data Fig. 9). Alteration of NCS clamp residues individually, and simultaneously, also abrogated the ability of Cas12a2 to clear target plasmid in vivo, providing direct evidence that aromatic clamp-mediated dsDNA duplex melting is essential for Cas12a2 activity. These data also suggest that dsDNA duplex degradation is the driving force behind Cas12a2-mediated immunity, as mutants that retained ssRNase and ssDNase activities were not sufficient to provide immunity.

Collectively these data show that Cas12a2 mediates abortive infection through a unique mechanism of dsDNA cleavage, distinct from indiscriminate RNA cleavage-induced abortive infection in other RNA-targeting CRISPR–Cas systems[13,30]. The catalytic mechanism of Cas12a2 is consistent with that of other RuvC endonucleases[21,31–33], for which protein-induced structural tension of the DNA facilitates proper scissile phosphate coordination. However, compared to the available RuvC-containing Cas9 and Cas12 structures, the NCS aromatic clamps provide a unique strategy to cleave duplexed nucleic acids.

## Discussion

Our results support a detailed mechanism of Cas12a2 in antiphage defence. Hybridization of the crRNA to PFS-containing RNA targets drives major conformational changes in Cas12a2, exposing the RuvC active site and alleviating autoinhibition (Fig. 4). In the active complex, the RuvC domain is located within an approximately 30-Å-wide positively charged groove that is sufficiently large to accommodate duplexed nucleic acids. Nonspecific electrostatic interactions facilitate collateral substrate capture, accompanied by duplex distortion and local base-pair melting. The melted bases are stabilized by two pairs of aromatic clamps that enable appropriate positioning of single nucleic acid strands within the RuvC active site. This multiple-turnover DNA nicking culminates in dsDNA degradation, distinct from the single-turnover dsDNA cleavage of Cas12a or Cas9 (ref. [16,34]) and enables robust and widespread DNA destruction in vivo.

Although Cas12a and Cas12a2 can use the same crRNA owing to their similar WED and RuvC domains, multiple structural divergences confer disparate target preferences and biochemical activities, and enable Cas12a2 to evade inhibition by multiple Cas12a-targeting anti-CRISPR proteins. Notably, the lack of a Nuc domain (also referred to as target loading domain, or target nucleic acid binding domain[35,36]) may contribute to the inability of Cas12a2 to directly bind dsDNA targets through the formation of an R-loop, conferring specificity for ssRNA target binding. Additionally, the lack of a Cas12a2 Nuc domain increases the accessibility to the RuvC nuclease active site, enabling rapid duplex capture and cleavage in *trans*. Previous structures of Cas12b and Cas12i bound to bystander ssDNA have shown that *trans* ssDNA substrates[22,23] must follow a tortuous, narrow path to reach the RuvC active site, which is incompatible with rigid dsDNA substrates (Extended Data Fig. 6). This is in contrast to the highly exposed Cas12a2 RuvC active site, providing a structural mechanism for dsDNA cleavage in *trans*.

While the mechanism of Cas12a2 is reminiscent of RNA targeting by Cas13, there are several distinct differences. Target binding by Cas13 activates nonspecific RNA degradation both in *cis* and in *trans*, resulting in persistent nuclease activation as the phage genome continues to produce target transcripts[30,37,38]. RNA cleavage can occur in *cis* if the target is of suitable length to fully hybridize with the crRNA and reach the distal active site opposite the seed region[39]. Similarly, Cas12a2 can trim target RNA in *cis*, but the hybridized target strand remains intact, enabling uninterrupted Cas12a2 activation (Extended Data Fig. 10). Cas13 systems therefore act as sentinels for viral RNA, whereas Cas12a2 represents the self-destruct button, conferring population-level antiphage defence through abortive infection. The consequences of Cas12a2 activation may be mitigated through protein degradation

and turnover coupled with the removal of phage transcript-encoding DNA by other defence systems (for example, restriction–modification systems and DNA-targeting CRISPR–Cas systems)[40].

Our discovery of the Cas12a2(Y1069A) point mutant that can cleave ssDNA but not dsDNA or ssRNA provides a blueprint for an RNA sensor with collateral activity that does not destroy the RNA target it has been programmed to detect or get distracted by the excess of other RNAs in the sample, which might enable the development of a highly sensitive RNA diagnostic.

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

## Methods

### Mutagenesis

Strains, plasmids, primers and nucleic acid substrates can be found in Supplementary Table 1. The mutant constructs used in this paper were prepared by point-directed mutagenesis using the Q5 Site-Directed Mutagenesis Kit (NEB). Primers were designed and annealing temperatures were determined using the NEBase Changer tool. All primers were ordered through IDT. Plasmid ligation was achieved with KLD Enzyme Mix (NEB) and completed plasmids were sequenced to verify correct alterations (Plasmidsaurus).

### Expression and purification

N-terminal hexa-his-tagged Cas12a2 and various mutant plasmids were transformed into chemically competent *Escherichia coli* NiCo21 cells (NEB). A single colony from transformation was selected for starter culture in 20 ml LB medium grown at 37 °C overnight (16–18 h). Each starter culture was used to inoculate 1.0 l of TB medium, and then cultures were grown up to an optical density at 600 nm of 0.6 at 37 °C. Cultures were cooled in ice for 15 min before being grown at 18 °C for a further 16–18 h followed by collection by centrifugation. Cell pellets were stored at −80 °C or used immediately for protein purification.

Cell pellets were resuspended in lysis buffer (25 mM Tris pH 7.2, 500 mM NaCl, 2 mM MgCl$_2$, 10 mM imidazole, 10% glycerol) treated with protease inhibitors (2 µg ml$^{-1}$ aprotinin, 10 µM leupeptin, 0.2 mM AEBSF, 1.0 µg ml$^{-1}$ pepstatin) and 1 mg ml$^{-1}$ lysozyme and incubated for 30 min on ice with shaking. Cells were lysed by sonication and clarified by centrifugation. Clarified lysate was batch-bound to nickel resin for 30 min at 4 °C and then allowed to flow through. Lysate was then passed again over nickel resin twice. Nickel resin was washed with nickel wash buffer (25 mM Tris pH 7.2, 2 M NaCl, 2 mM MgCl$_2$, 10 mM imidazole, 10% glycerol) and eluted with nickel elution buffer (25 mM Tris pH 7.2, 500 mM NaCl, 2 mM MgCl$_2$, 250 mM imidazole, 10% glycerol). Nickel elutions were desalted into low-salt buffer (25 mM Tris pH 7.2, 50 mM NaCl, 2 mM MgCl$_2$, 10% glycerol) using either HiPrep 26/10 desalting column (Cytiva) or Pd10 Sephadex G25M Columns (Cytiva) depending on purification size. Protein samples were then loaded over an ion-exchange column (HiTrap SP HP or HiTrap Q HP column (Cytiva) depending on the isoelectric point of the construct) using low-salt buffer and eluted with a gradient of high-salt buffer (25 mM Tris pH 7.2, 1.0 M NaCl, 2 mM MgCl$_2$, 10% glycerol).

Peak elutions were then pooled and concentrated to about 1.0 ml. During this process, protein to be used for biochemistry was desalted by refilling the concentrator with low-salt buffer twice to exchange out the high-salt buffer. Concentrated protein for cryo-EM was loaded over a HiLoad 26/600 Superdex 200 pg column (GE Healthcare) using SEC buffer (25 mM HEPES pH 7.2, 150 mM NaCl, 2 mM MgCl$_2$, 5% glycerol). Peak fractions were then pooled and concentrated again. Concentrated protein was either flash frozen in liquid nitrogen or used in complex formation for cryo-EM.

### Complex formation for cryo-EM

Before crRNA was added to Cas12a2, RNA was incubated at 65 °C for 3 min followed by cooling 1 °C min$^{-1}$ to room temperature. Binary complex was formed for cryo-EM by combining protein and synthetic crRNA in a 1:1.2 molar ratio in SEC buffer (25 mM HEPES pH 7.2, 150 mM NaCl, 2 mM MgCl$_2$, 5% glycerol) and incubating at 24 °C for 10 min. Unbound crRNA was then separated from the binary complex over a Superdex 200 10/300 increase GL sizing column (Cytiva) into Cryo-EM Buffer (12.5 mM HEPES pH 7.2, 150 mM NaCl, 2 mM MgCl$_2$). Eluted protein was concentrated to 30 µM in a 100-kDa MWCO spin concentrator (Corning) and flash frozen in liquid nitrogen.

### Far-UV circular dichroism spectroscopy

Protein samples of Cas12a2 mutants were prepared at a concentration in the range 0.3–0.5 mg ml$^{-1}$ determined by nanodrop in circular dichroism (CD) buffer (20 mM Tris pH 7.2, 100 mM NaCl). Far-UV CD readings used a Jasco-J1500 spectropolarimeter. The CD spectra were obtained from 260–190 nm using a scanning speed of 50 nm min$^{-1}$ (with a 2 s response time and accumulation of three scans). Melting curves of Cas12a2(ΔPI) samples (in sealed quartz cuvettes with 0.1 cm path length) were obtained by monitoring the CD signal at 222 nm every 1 °C over a 10–90 °C temperature range, using a temperature ramp of 15 °C h$^{-1}$. The CD signal was converted to molar ellipticity by Jasco Spectra Manager software.

### Plasmid curing assay

Plasmid curing assays were conducted as described previously[1]. In short, immune system plasmids were prepared with Cas12a2 and a 3× CRISPR repeat and transformed into BL21 AI cells by heat-shock transformation. Cells expressing the immune system were then made electrocompetent[41] and immediately transformed by electroporation with 50 ng of either target or non-target plasmid. Transformations were recovered for 18 h in 450 µl LB medium containing 1 mM IPTG, 0.2% L-arabinose and antibiotics for the immune system plasmid. Recovered transformations were then serially diluted in LB medium between 10$^1$ and 10$^6$ and spotted in 10-µl drops on LB agar plates containing 1 mM IPTG, 0.2% L-arabinose and antibiotics for both immune system and target or non-target plasmids. Colonies were counted in the highest countable spot in the dilution series and the relative transformation efficiency was calculated between the target and non-target plasmid.

### Activation assay

Binary complex of either WT Cas12a2 or Cas12a2(ΔPI) with crRNA was combined with various targets (FL, ΔPFS, Δ5 and Δ10) to final reaction conditions of 600 nM Cas12a2, 720 nM crRNA and 300 nM FAM-labelled target in 1× NEB 3.1 Buffer (50 mM Tris pH 7.9, 100 mM NaCl, 10 mM MgCl$_2$, 100 µg ml$^{-1}$ BSA). Binary complex was first formed by combining WT Cas12a2 or Cas12a2(ΔPI) with crRNA in a 1:1.2 molar ratio and incubating for 30 min at room temperature with NEB 3.1 buffer as a 2× master mix. Binary complex and target RNAs were then combined to their final reaction concentration and incubated at room temperature for 1.0 h before quenching with 1:1 (v/v) phenol–chloroform pH 4.5 mixed by flicking.

Reactions were quenched with phenol–chloroform pH 4.5 and mixed by flicking followed by spinning down for 30 s. Reaction products were run on a 12% fully denaturing formaldehyde (FDF) polyacrylamide gel electrophoresis (PAGE) gel as described by previously[42] with minor modifications. Loading dye was replaced with 30% glycerol, and gels were run at 50 V for 15 min before increasing the voltage to 150 V.

### Target protection assay

Binary complex of Cas12a2 and crRNA was combined with FAM-labelled FL target RNA to a final reaction condition of 600 nM Cas12a2, 720 nM crRNA, 300 nM target RNA in 1× NEB 3.1 buffer (50 mM Tris pH 7.9, 100 mM NaCl, 10 mM MgCl$_2$, 100 µg ml$^{-1}$ BSA). Binary complex was first formed as described in the activation assay. The 2× master mix was then combined with the target RNA and incubated at 37 °C. Samples were taken at time points 5, 15, 30, 60 and 120 min, and quenched in pH 4.5 phenol–chloroform. Samples were then run on a 12% FDF–PAGE gel as described in the activation assay. Completed gels were imaged for FAM fluorescence and then stained with SYBR Gold and imaged again to show unlabelled RNA species.

### Effect of ratios on cleavage

FAM-labelled FL target RNA (300 nM) was combined with a range of binary complex concentrations (600, 300, 150, 75 and 37.5 nM) to achieve complex/target ratios of 2:1, 1:1, 1:2, 1:4 and 1:8. Binary complex was first formed by combining Cas12a2 (1,200 nM) with crRNA in a 1:1.2 molar ratio and incubating for 30 min at 37 °C with NEB 3.1 buffer as a 2× master mix. Binary complex was then serially diluted in 2× NEB 3.1 buffer

(100 mM Tris pH 7.9, 200 mM NaCl, 20 mM MgCl$_2$, 200 µg ml$^{-1}$ BSA) to form a 2× master mix for each complex/target ratio. Each 2× master mix was combined with target RNA and incubated at 37 °C for 1.0 h followed by quenching with phenol–chloroform pH 4.5. Quenched samples were visualized as described in the activation assay.

### Trans-cleavage assay

Reactions containing 600 nM Cas12a2 (WT or mutants) and 720 nM crRNA were combined with 300 nM FL target RNA and 300 nM FAM-labelled non-target RNA, ssDNA or dsDNA. Binary complex was first formed as described previously as a 4× master mix. The master mix was combined with target and non-target substrates and incubated at 37 °C for 1.0 h and quenched with phenol–chloroform pH 4.5. Samples were then separated on a 12% 7 M urea PAGE gel and imaged for FAM fluorescence.

To test the effect of pre-incubation of target with Cas12a2, 100 nM Cas12a2–crRNA complex was incubated for 2 h at room temperature with 200 nM target RNA in NEB 3.1 buffer (50 mM Tris-HCl pH 7.9, 100 mM NaCl, 10 mM MgCl$_2$, 100 µg ml$^{-1}$ BSA). After incubation, Cas12a2/crRNA/target mixture was combined with 1 µM RNAse Alert or DNAse Alert (IDT), and the reactions were allowed to proceed for 60 min at room temperature. These reactions were compared to the same reaction condition with target RNA added simultaneously (no incubation) with RNAse Alert or DNAse Alert. Fluorescent signal was tracked with a Synergy H4 plate reader (BioTek) and data were plotted in GraphPad Prism.

### Plasmid cleavage assay

Plasmid cleavage reactions were prepared by combining 14 nM Cas12a2 (or mutants) with 14 nM crRNA and 25 nM target RNA in NEB 3.1 buffer (50 mM Tris-HCl pH 7.9, 100 mM NaCl, 10 mM MgCl$_2$, 100 µg ml$^{-1}$ BSA). Protein was preheated at 37 °C for 15 min before the addition of 7 nM supercoiled pUC19 plasmid. Samples were taken at time points of 1, 2, 5, 10, 20, 30, 45 and 60 min and quenched in pH 8.0 phenol–chloroform. Quenched reactions were mixed by flicking followed by centrifugation. Samples were loaded on 1% agarose gels and visualized with ethidium bromide.

### Cryo-EM sample preparation and data acquisition and processing

Flash-frozen Cas12a2 binary complex was rapidly thawed. A 4 µl volume of the binary complex was applied to C-flat holey carbon grids (2/2, 400 mesh) that had been plasma-cleaned for 30 s in a Solarus 950 plasma cleaner (Gatan) with a 4:1 ratio of O$_2$/H$_2$. Grids were blotted with Vitrobot Mark IV (Thermo Fisher) for 2 s, blot force 4 at 4 °C and 100% humidity, and plunge-frozen in liquid ethane. Data were collected on an FEI Glacios cryo-TEM equipped with a Falcon 4 detector. Data were collected in SerialEM v3.8, with a pixel size of 0.94 Å, a defocus range of −1.5 to −2.5 µm and a total exposure time of 15 s resulting in a total accumulated dose of 40 electrons Å$^{-2}$ that was split into 60 electron event representation fractions. Motion correction, contrast transfer function (CTF) estimation and particle picking was carried out on-the-fly using cryoSPARC Live v4.0.0-privatebeta.2 (ref. [43]). A total of 1,577 videos were collected, of which 1,159 were accepted on the basis of meeting the criteria of a CTF fit of 5 Å or better. All subsequent data processing was carried out in cryoSPARC v3.2 (ref. [44]).

From 987,122 particles picked, 214,647 were selected from a single round of two-dimensional (2D) classification. These particles were subjected to ab initio reconstruction (three classes) followed by heterogeneous refinement resulting in a final subset of 97,470 particles that yielded a 3.46-Å-resolution structure from non-uniform refinement. Re-extraction of this subset of particles in a 320-pixel box size, splitting of particles into 4 exposure groups and carrying out per-group CTF refinement and per-particle defocus optimization as implemented in non-uniform refinement[45] resulted in a 3.2-Å-resolution reconstruction that was used for modelling.

For the ternary complex, a rapidly thawed Cas12a2 binary complex fraction was supplemented with a fourfold excess of heat-annealed (90 °C for 5 min, and rapidly cooled to 4 °C) PFS-containing RNA target and incubated at room temperature (about 25 °C) for 30 min before vitrification, which was carried out in an identical manner to that for the binary complex as described above. Data were collected using an FEI Titan Krios cryo-electron microscope equipped with a K3 Summit direct electron detector (Gatan). Images were recorded with SerialEM[46] with a pixel size of 0.81 Å. A total accumulated dose of 70 electrons Å$^{-2}$ during a 6-s exposure was fractionated into 80 frames. A total of 6,940 micrographs were collected, of which 6,614 with CTF fits of 5 Å or better were retained. On-the-fly processing was carried out as described above.

A total of 3,515,037 particles were picked, of which 2,212,319 were selected after 2D classification. Multiple rounds of ab initio reconstruction and heterogeneous refinement resulted in a subset of 192,639 particles that were reconstructed to 2.92-Å-resolution using non-uniform refinement as described above. This map was then used for modelling.

For the quaternary complex, ternary complex was prepared as described above, and incubated with heat-annealed phosphothioate dsDNA duplex for 30 min at room temperature. A 2.5 µl volume of complex was applied to C-flat grids (1.2/1.3, 300 mesh) and blotted for 6 s, blot force 0 at 4 °C and 100% humidity before vitrification. Data were collected on an FEI Glacios cryo-TEM equipped with a Falcon 4 detector, as described for the binary complex. A total of 1,755 videos were collected, of which 1,539 had CTF fits of 5 Å or better and were retained for subsequent processing. On-the-fly motion correction, CTF estimation and particle picking were carried out as described above.

A total of 1,692,368 particles were picked, of which 425,770 were retained after a single round of 2D classification. A single round of ab initio reconstruction (3 classes) followed by heterogeneous refinement yielded a subset of 260,958 particles that were reconstructed to 2.97 Å resolution using non-uniform refinement. Additional rounds of ab initio reconstruction and heterogeneous refinement were used to further classify particles, resulting in a final subset of 104,857 particles. Extraction of said particles with a 384-pixel box, splitting particles into 9 exposure groups, and reconstruction using non-uniform refinement with per-group CTF refinement and per-particle defocus optimization resulted in a 2.74-Å-resolution reconstruction that was then used for modelling.

### Model building and figure preparation

A Cas12a binary complex (Protein Data Bank (PDB) 5NG6)[20] was rigid body fitted into the Cas12a2 binary complex map. Although most of the model did not correspond to the map, the RuvC and WED domains generally were consistent. However, pairwise blast of the two proteins revealed multiple gaps or inserts for the relative insertions. However, a single 20-residue hairpin of the Cas12a RuvC domain fitted the Cas12a2 map well and had no gaps or insertions in the pairwise blast. This fragment was isolated and rigid body fitted into the Cas12a2 map, and the sequence was mutated to the corresponding region of Cas12a2 using Coot v1.0 (ref. [47]). This was then used as a fiducial to build the rest of the complex de novo using in Coot. Attempts at using AlphaFold2 (AF2)[48] to generate fragments to fit in the map were unsuccessful as adjacent residues within the WED and RuvC domains were separated by protein sequence and the REC1 and REC2 domain boundaries were not obvious from the sequence alone. However, AF2 was used to validate modelling of small structural domains after-the-fact, for which smaller, compact regions of the model were folded using AF2, and fitted into the map, indicating correct modelling. This was particularly useful when the de novo model contained gaps due to local flexibility.

The 5′ crRNA handle was built using the Cas12a 5′ crRNA as a template (PDB 5NG6). In the binary complex, the seven-nucleotide 3′ seed region was modelled de novo as polyU as it was not possible to unambiguously determine nucleotide identity.

Once fully modelled, Isolde v1.4 (ref. [49]) was used to improve the fit of the model to the map, and real-space refinement as implemented within Phenix v1.19 (ref. [50]) was carried out to optimize model geometry.

For the Cas12a2 ternary complex, the RuvC, WED, and part of the insert domains were in the same conformation as in the binary complex. These were rigid body fitted into the ternary complex map. The REC1, REC2 and the C-terminal half of the insertion domain were separately fitted as rigid bodies into the ternary complex map. The PI domain structure was predicted using AF2, and then manually connected to the rest of the model. The crRNA–target RNA duplex was modelled as ideal A-form RNA within Coot, and manually connected to the 5′ crRNA handle. The target RNA 3′ PFS was modelled de novo. Coot was used to fit in gaps within the model, and Isolde was then used to improve the quality of model before real-space refinement as described above.

For the quaternary complex, the ternary complex structure was rigid body fitted into the map, and then flexibly fitted using Isolde. The ZR domain structure was predicted using AF2, and manually connected to the rest of the model. The dsDNA duplex was modelled de novo, with one strand modelled as polyT and the other as polyA as it was not possible to unambiguously determine nucleotide identity. $Mg^{2+}$ and $Zn^{2+}$ ions and an activating $H_2O$ were modelled manually using the sharpened map. Isolde and real-space refinement were carried out as described above.

All structural figures and videos were generated using ChimeraX v1.0 (refs. [51,52]), apart from the modevectors, which were generated in PyMol v2.5.

## Reporting summary

Further information on research design is available in the Nature Portfolio Reporting Summary linked to this article.

## Data availability

The atomic models of Cas12a2 binary, tertiary and quaternary complexes have been deposited into the PDB with accession codes 8D49, 8D48 and 8D4A, and the corresponding maps have been deposited into the Electron Microscopy Data Bank with codes EMD-29178, EMD-27180 and EMD-27179, respectively. Source data are provided with this paper.

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

**Acknowledgements** We thank members of the laboratories of R.N.J. and D.W.T. for helpful discussions, A. Brilot for expert cryo-EM technical assistance and O. Dmytrenko for initial guidance on the in vivo plasmid interference assays. Data were collected at the Sauer Structural Biology Laboratory at the University of Texas at Austin. This work was supported in part by National Institute of General Medical Sciences of the National Institutes of Health R35GM138348 (to D.W.T.) and R35GM138080 (to R.N.J.), an ERC Consolidator grant (865973 to C.L.B.), the Federal Agency for Disruptive Innovation (to C.L.B.), Welch Foundation Research Grant F-1938 (to D.W.T.) and a Robert J. Kleberg, Jr. and Helen C. Kleberg Foundation Medical Research Grant (to D.W.T.). D.W.T is a CPRIT Scholar supported by the Cancer Prevention and Research Institute of Texas (RR160088) and an American Cancer Society Research Scholar supported by the American Cancer Society (RSG-21-050-01-DMC).

**Author contributions** J.P.K.B. carried out cryo-EM structure determination, model building and structural analysis. T.H. and B.N. purified and reconstituted the enzyme complexes. T.H. carried out biochemical and in vivo experiments. C.L.B. aided with in vivo experiments. J.P.K.B., T.H., R.N.J. and D.W.T. analysed and interpreted the data and wrote the manuscript. R.N.J. and D.W.T. supervised the study and obtained financial support for the work.

**Competing interests** J.P.K.B, T.H., R.N.J. and D.W.T. are inventors on a patent application based on this research titled 'Compositions and methods related to modified Cas12a2 molecules' filed by the Board of Regents, The University of Texas System. The US Patent and Trademark Office (USPTO) has assigned US application no. 63/349,225 to this application, and the filing date of 6 June 2022.

**Additional information**
**Correspondence and requests for materials** should be addressed to Ryan N. Jackson or David W. Taylor.

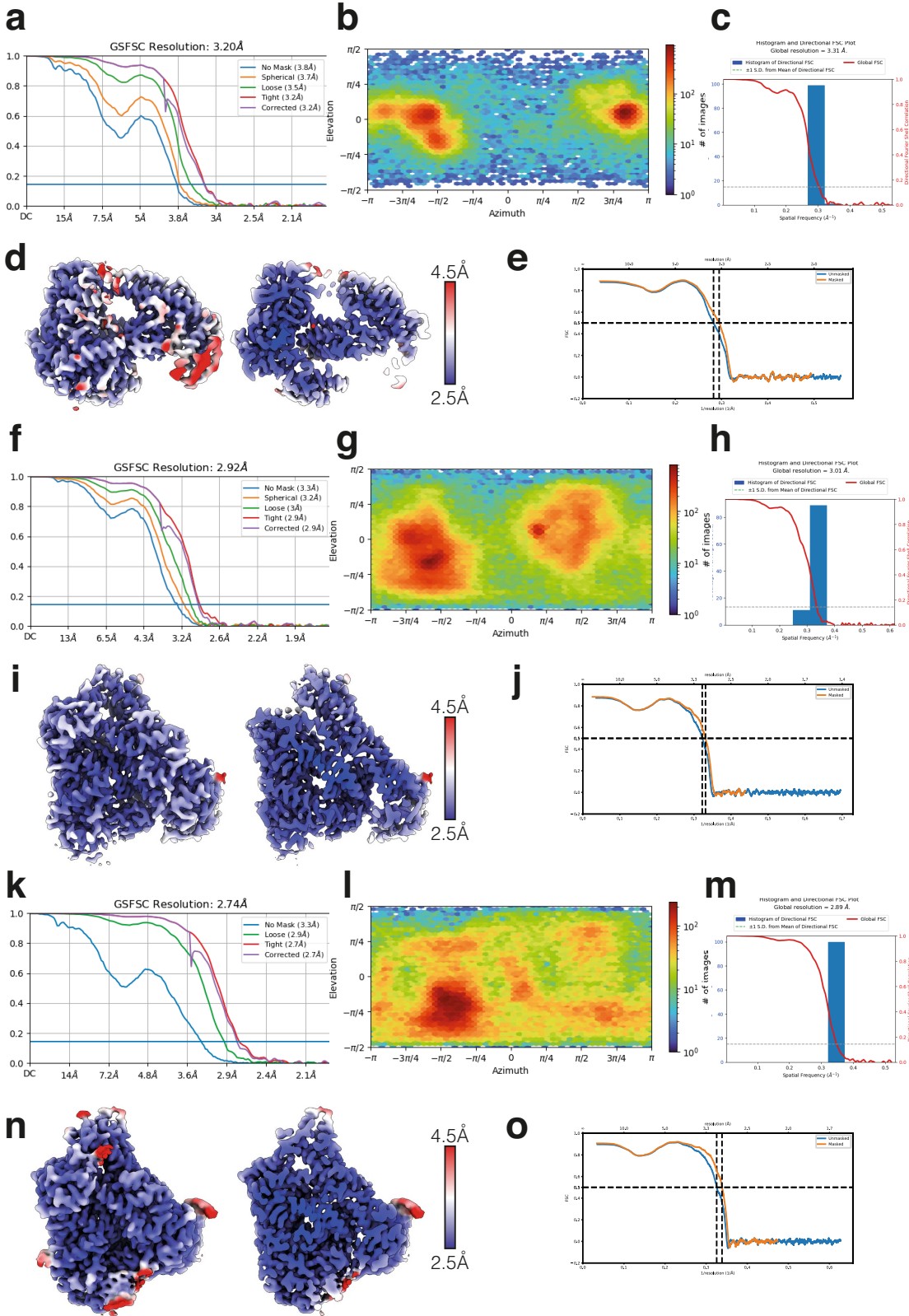

**Extended Data Fig. 1 | Cryo-EM data analysis. a**, Gold standard Fourier Shell Correlation (FSC) curves for Cas12a2 binary complex. **b**, Euler plot for Cas12a2 binary complex. **c**, Directional FSC[53] for Cas12a2 binary complex. **d**, Cas12a2 binary complex colored by local resolution. Left – surface, right, cut-through. **e**, Map-to-model FSC for Cas12a2 binary complex. **f**, Gold standard FSC for Cas12a2 ternary complex. **g**, Euler plot for Cas12a2 ternary complex. **h**, Directional FSC for Cas12a2 ternary complex. **i**, Cas12a2 ternary complex colored by local resolution. Left – surface, right, cut-through. **j**, Map-to-model FSC for Cas12a2 ternary complex. **k**, Gold standard FSC for Cas12a2 quaternary complex. **l**, Euler plot for Cas12a2 quaternary complex. **m**, Directional FSC for Cas12a2 quaternary complex. **n**, Cas12a2 quaternary complex colored by local resolution. Left – surface, right, cut-through. **o**, Map-to-model FSC for Cas12a2 quaternary complex.

## Cas12a2

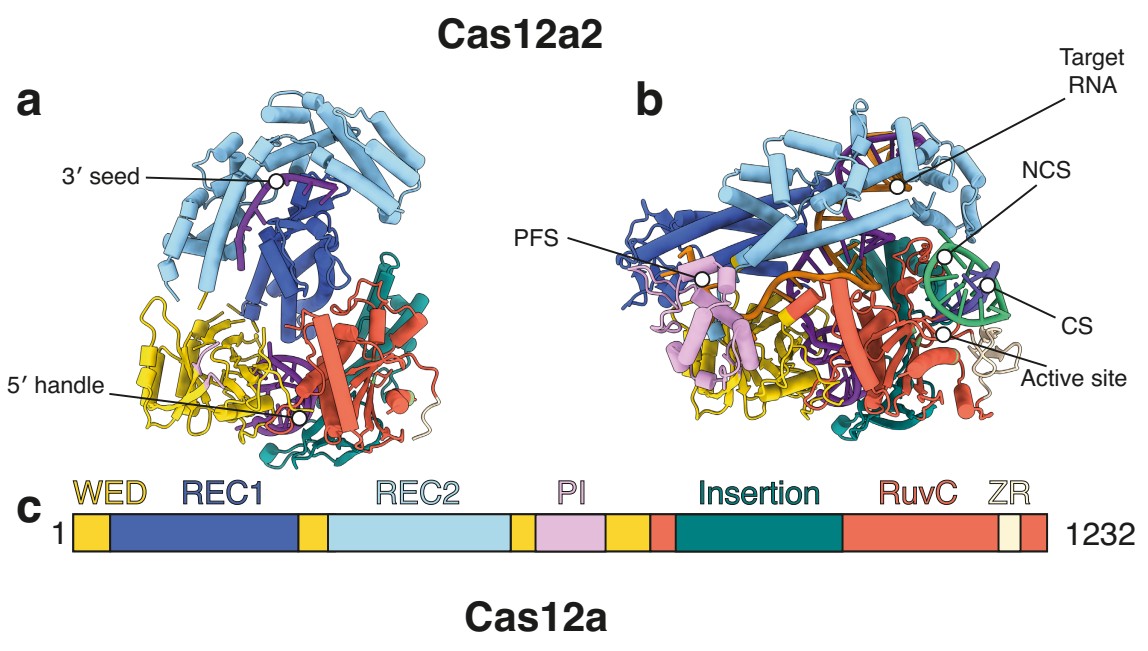

**a**

3′ seed

5′ handle

**b**

Target RNA

NCS

PFS

CS

Active site

**c**  WED  REC1  REC2  PI  Insertion  RuvC  ZR

1  1232

## Cas12a

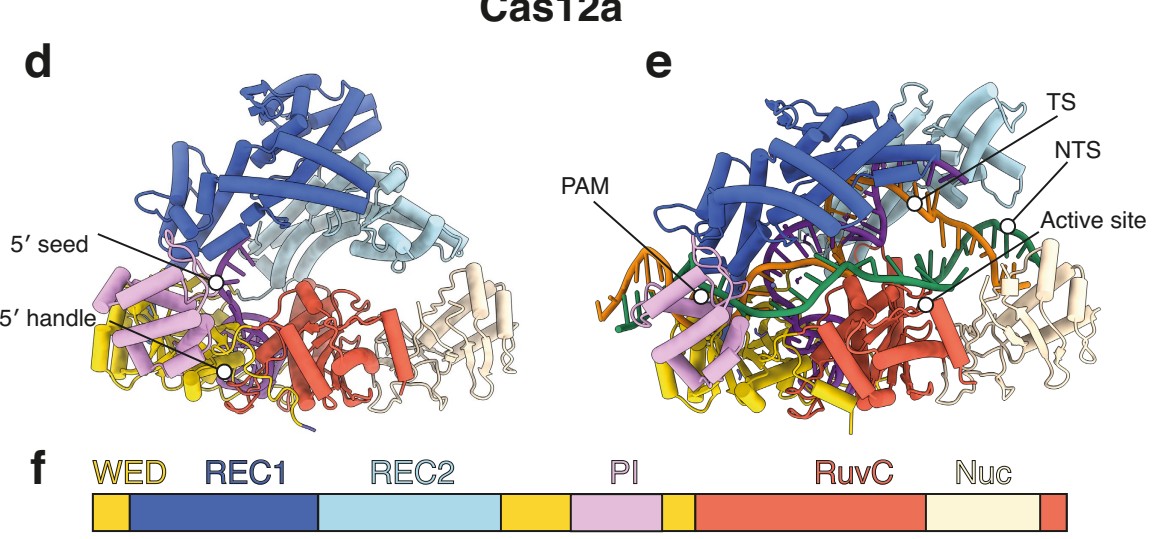

**d**

5′ seed

5′ handle

**e**

PAM

TS

NTS

Active site

**f**  WED  REC1  REC2  PI  RuvC  Nuc

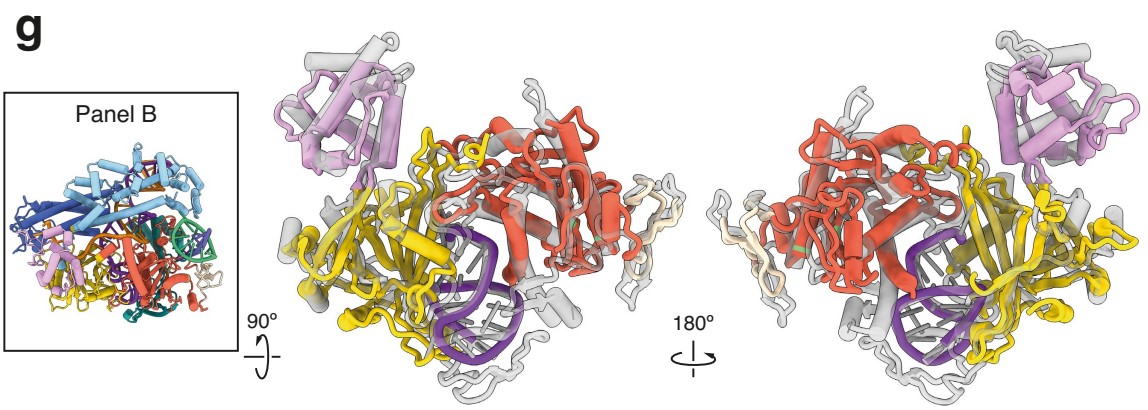

**g**

Panel B

90°

180°

**Extended Data Fig. 2 | Comparison of Cas12a2 and Cas12a. a‑c**, Cas12a2 binary complex (**a**), active quaternary complex (**b**) colored by structural domain (**c**). **d‑f**, Cas12a binary complex (**d**), active ternary complex (**e**) colored by structural domain (**f**). **g**, Overlay of 5′ crRNA handle, WED, RuvC and PI domains of Cas12a2 (colored) and Cas12a (transparent grey).

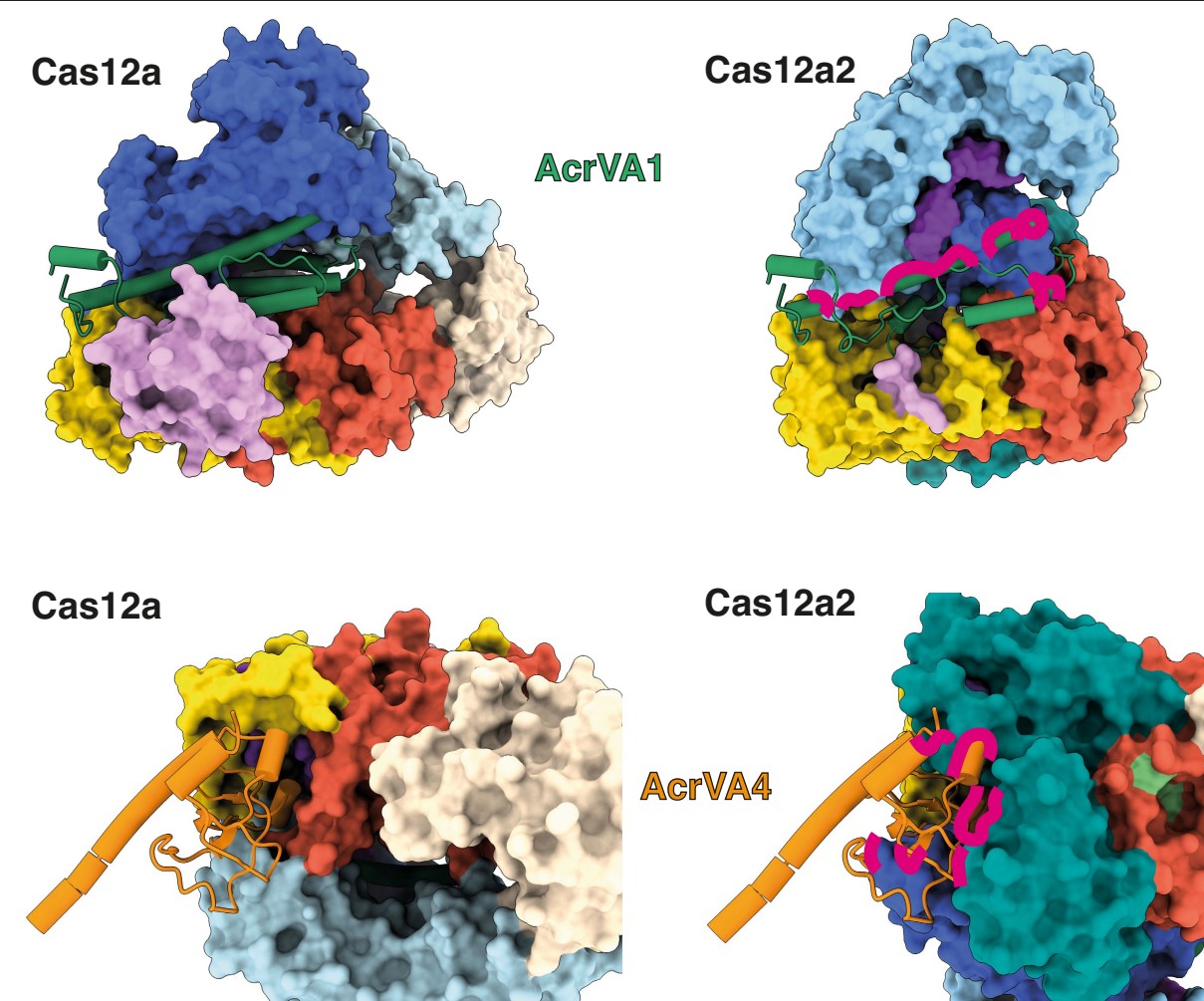

**Extended Data Fig. 3 | Structural basis for Anti-CRISPR evasion by Cas12a2.**
Top – Cas12a binary complex (PDB 5NG6, left) and Cas12a2 (right) with AcrVA1 (PDB 6NMD, green). AcrVA1 interacts with the 5' crRNA seed in Cas12a, which is not present in Cas12a2. There are also significant clashes between AcrVA1 and the REC lobe of Cas12a2. Middle – Cas12a (left) and Cas12a2 (right) binary complex with a monomer of AcrVA4 (PDB 6NMA, orange). There are severe clashes between the Cas12a2 Insertion domain (dark cyan) and AcrVA4. Clashes are shown as magenta lines.

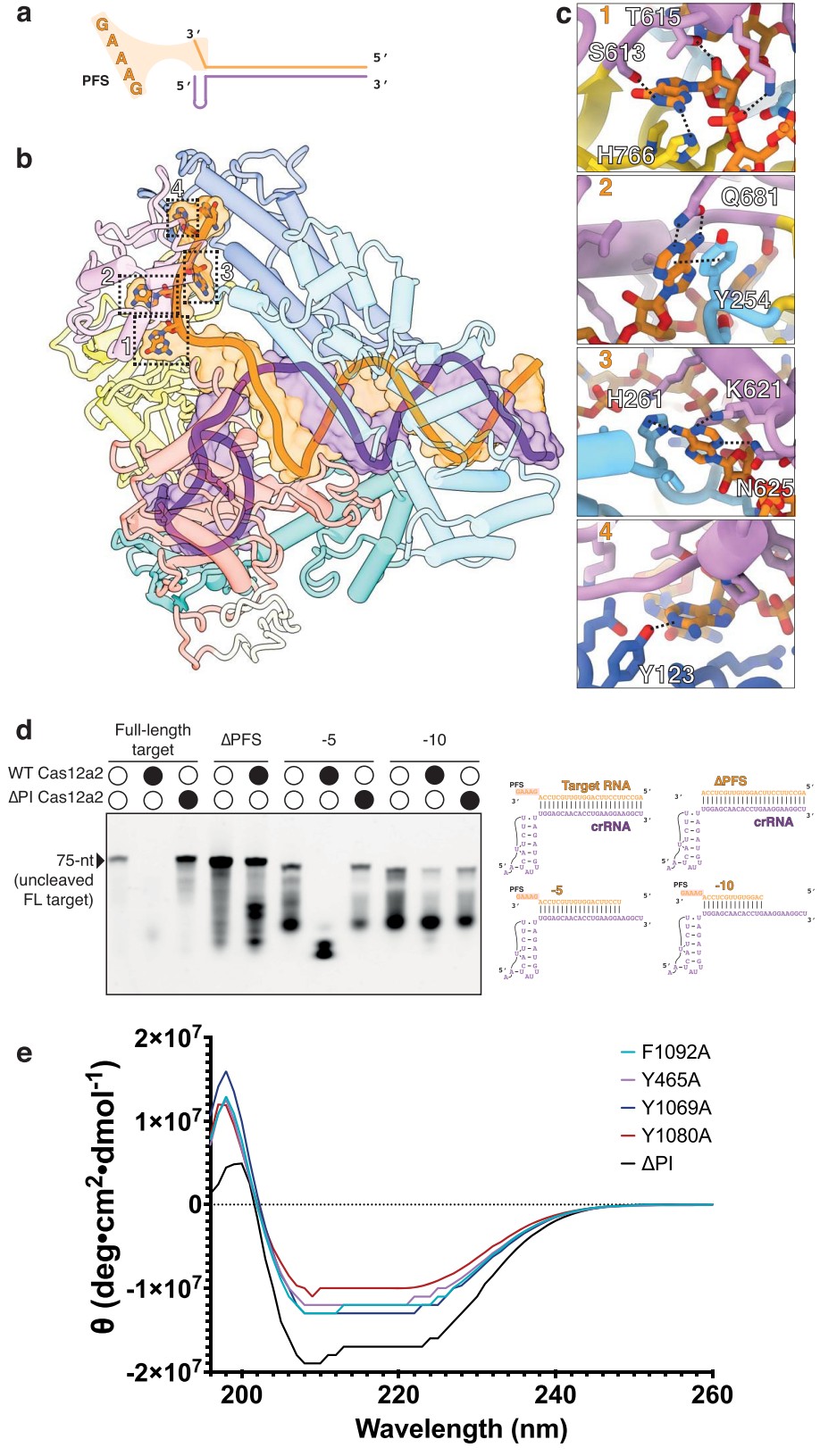

**Extended Data Fig. 4 | Mechanism of protospacer-flanking sequence (PFS) recognition by Cas12a2. a**, Schematic representation of PFS (GAAAG) located at the 3′ end of the RNA target RNA. **b**, Positions of the PFS within the Cas12a2 active ternary complex. **c**, Zoom-in of interactions between Cas12a2 and the first four bases of PFS. **d**, Activation of Cas12a2 or Cas12a2 ΔPI cleavage by truncated target RNAs. Representative of three independent experiments with similar results. **e**, Circular Dichroism (CD) spectroscopy of Cas12a2 mutants, including ΔPI truncation. All mutants are properly folded. For gel source data, see Supplementary Fig. 1.

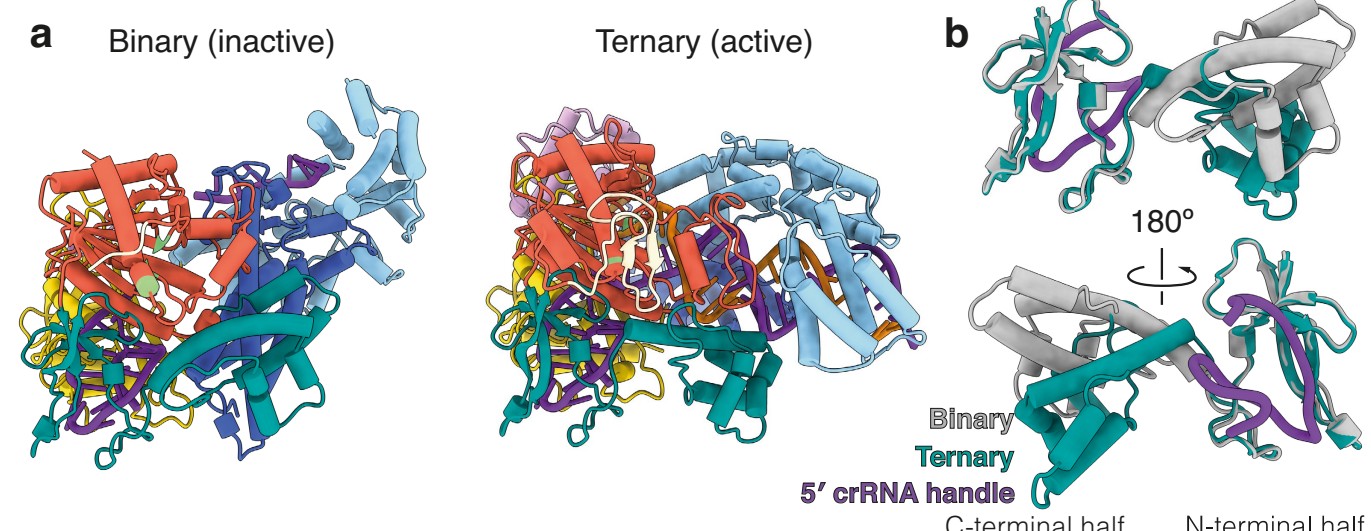

**a** Binary (inactive)   Ternary (active)

**b**

180°

Binary
Ternary
5′ crRNA handle

C-terminal half   N-terminal half

**Extended Data Fig. 5 | Conformational changes in Insertion domain upon target RNA binding. a**, Cas12a2 autoinhibited binary complex (left) and active ternary complex (right), colored as in Fig. 1a. Insertion domain is dark cyan.

**b**, conformational changes of the C-terminal half of insertion domain from binary (grey) to ternary (dark cyan) complexes. N-terminal half remains static, and interacts with the crRNA 5′ handle.

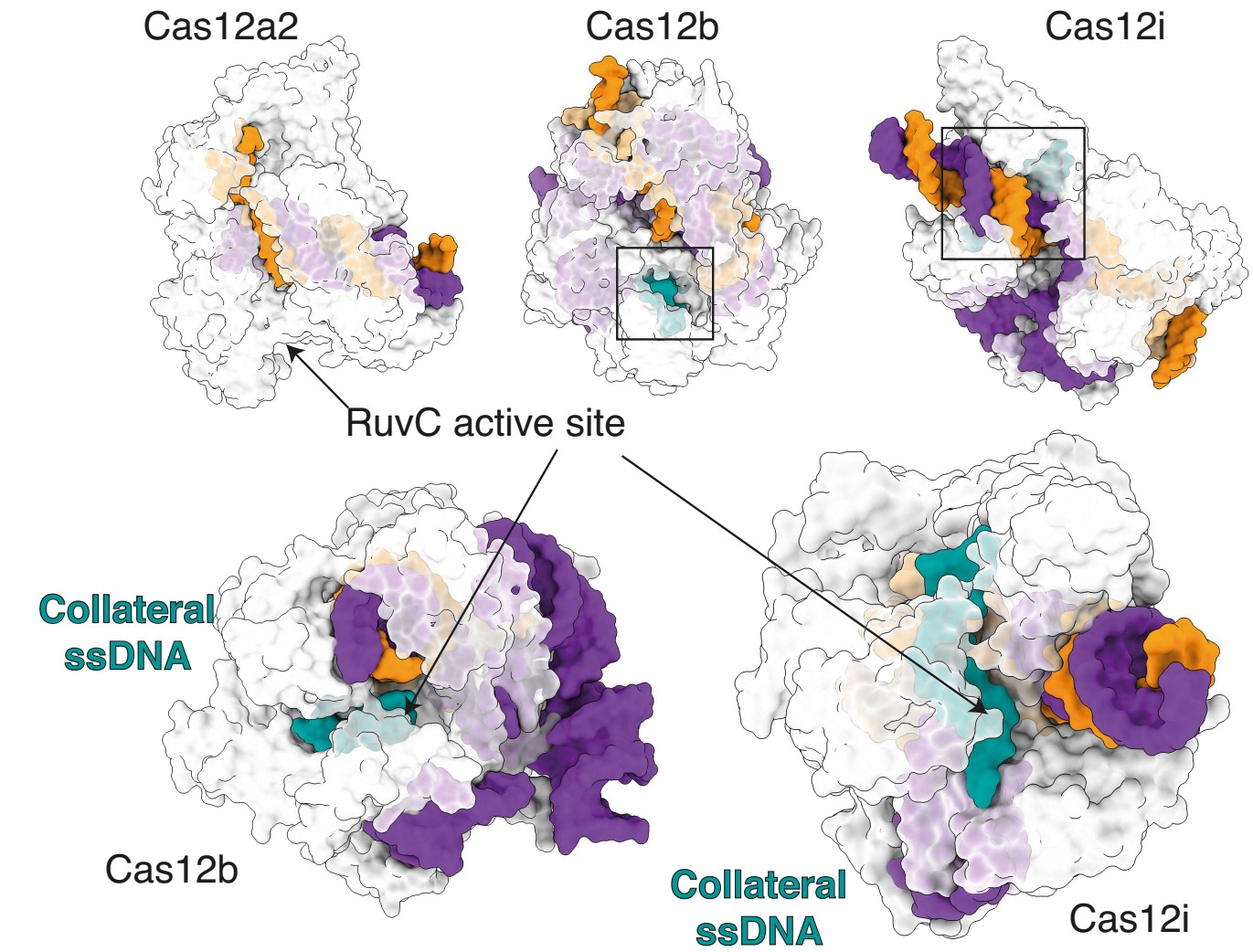

**Extended Data Fig. 6 | Collateral ssDNA in other Cas12 structures. Top**, Cas12a2 ternary complex (this study), Cas12b (PDB 5U31) and Cas12i (6W5C). Arrow denotes Cas12a2 active site. Boxes show position of bound collateral ssDNA in Cas12b and Cas12i. **Bottom**, Zoom-in of collateral ssDNA (dark cyan) buried in RuvC active site of Cas12b and Cas12i.

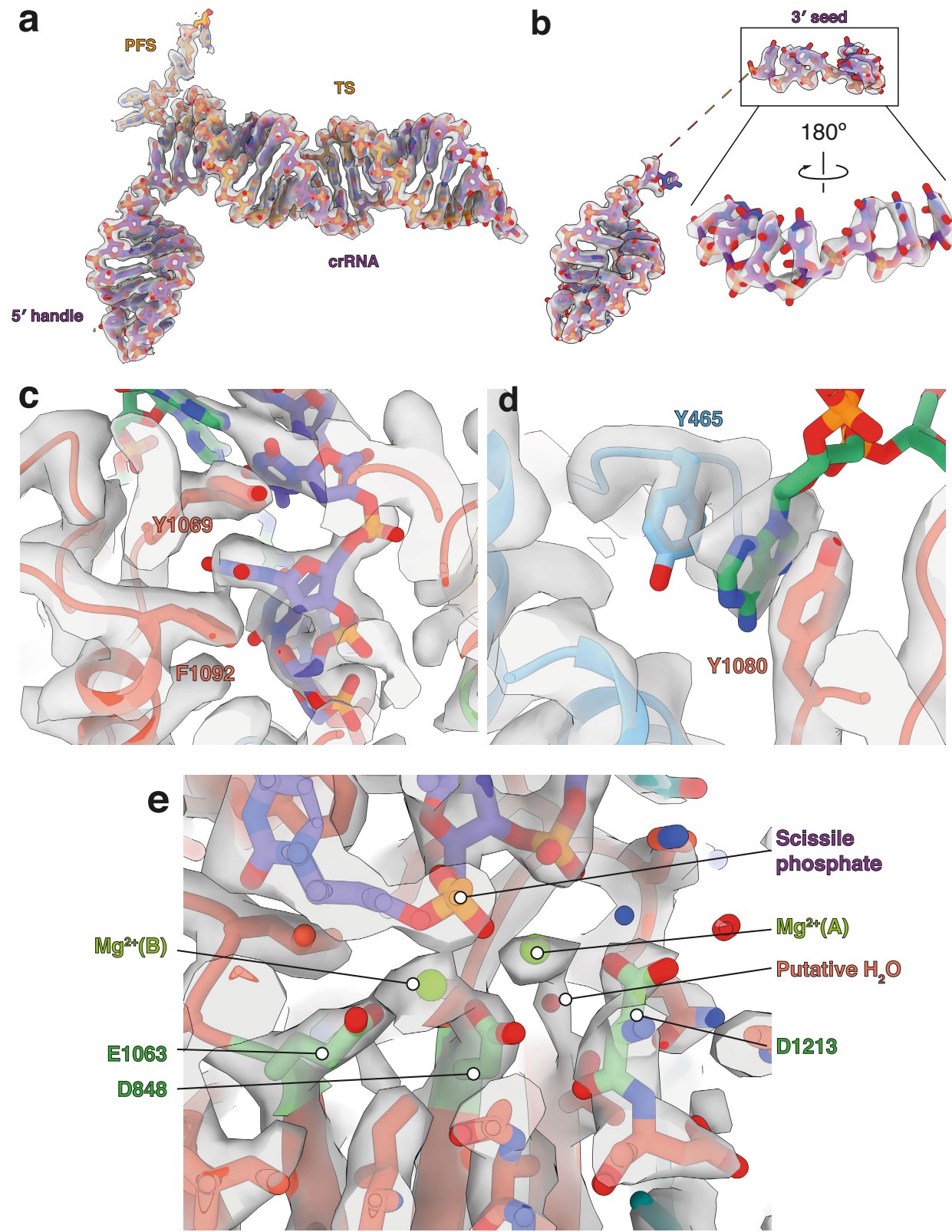

**Extended Data Fig. 7 | Representative cryo-EM densities. a, target RNA** and crRNA from ternary complex map, showing 5′ crRNA handle and 3′ target RNA PFS. **b**, 5′ handle and 7-nt pre-ordered seed from Cas12a2 binary complex. **c**, Cleaved strand pair of aromatic clamps. **d**, non-cleaved strand (NCS) aromatic clamps. Density for the unwound bases of the NCS is diffuse due to flexibility, but the nucleobase held within the aromatic clamp is well-resolved. **e**, RuvC active site, including density for scissile phosphate, two $Mg^{2+}$ ions and a putative activating water.

## a

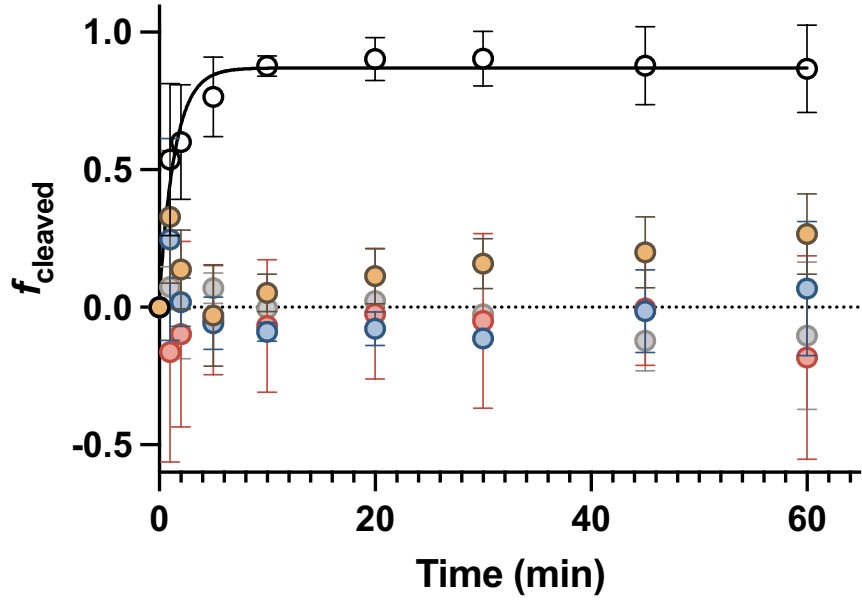

### Supercoiled Plasmid Cleavage

— WT Cas12a2
● Cas12a2 Y465A
● Cas12a2 Y1069A
● Cas12a2 Y1080A
● Cas12a2 F1092A

## b

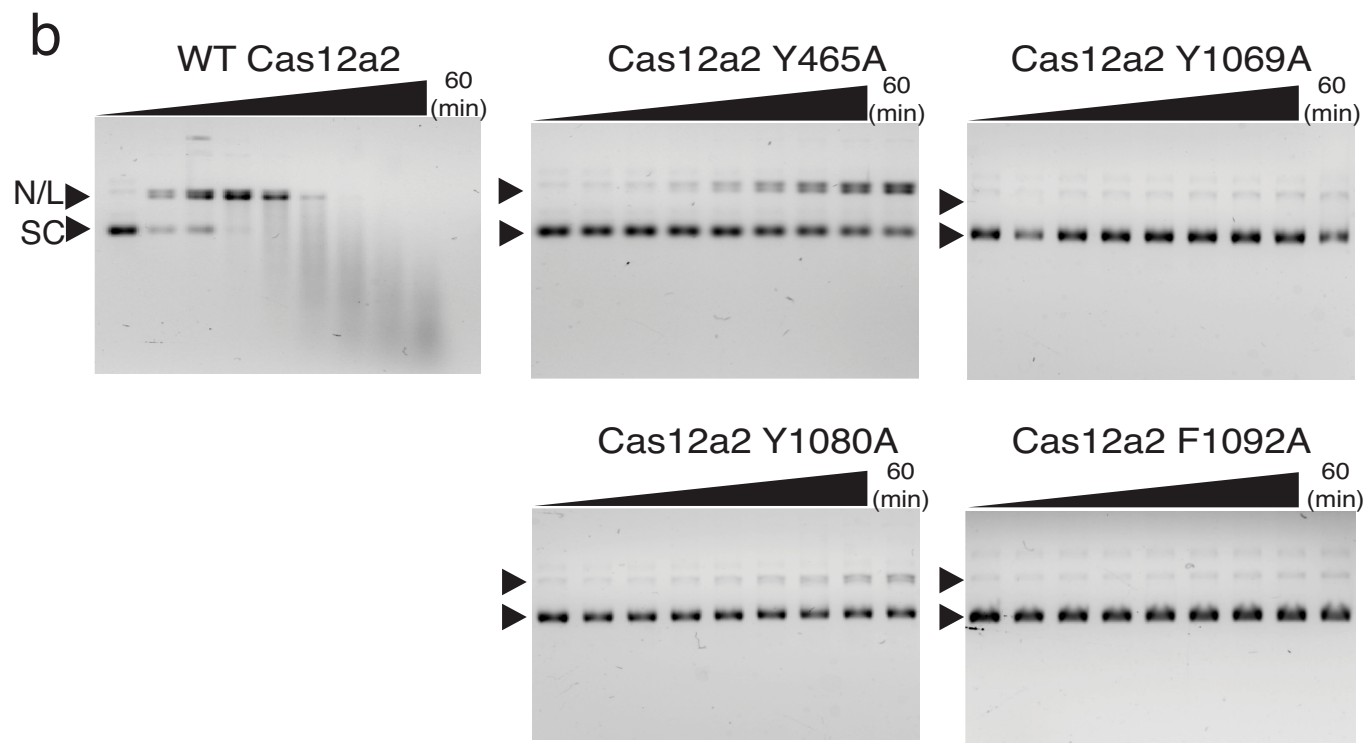

**Extended Data Fig. 8 | Effect of clamp residue mutations on supercoiled DNA cleavage *in vitro*.** Rate of supercoiled DNA cleavage by wild-type Cas12a2 and the four aromatic clamp mutants described in Fig. 3. WT Cas12a2 completely cleaves supercoiled plasmid within 10 min, while all four mutants significantly reduce complete cleavage. Y465A retains an ability (albeit much slower than WT Cas12a2) to nick and/or linearize plasmid, but fails to fully degrade plasmid DNA. **a**, Quantitation of fraction of supercoiled pUC19 cleaved over time. Significance between WT and mutant SuCas12a2 was determined by two-sided Student T-test. *P < 0.05, **P < 0.01, ***P < 0.001. Experiments were performed in triplicate, and error bars correspond to the mean and standard error. **b**. Representative agarose gels of supercoiled pUC19 cleavage by WT Cas12a2 or aromatic clamp mutants over time. Gels are representative of three independent experiments. For gel source data, see Supplementary Fig. 1.

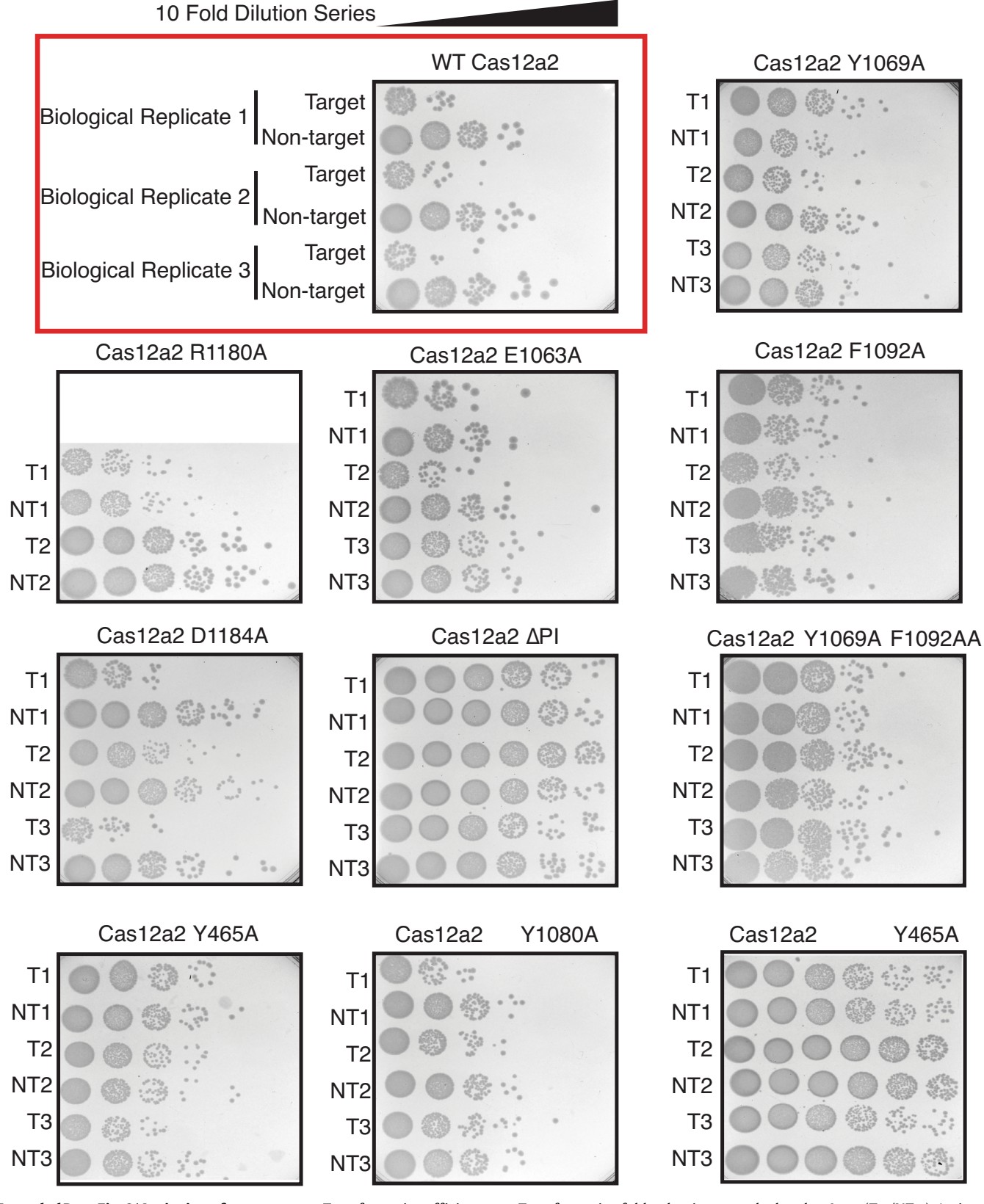

**Extended Data Fig. 9 | *In vivo* interference assays.** Transformation efficiency of Target (T) and Non-Target (NT) plasmids into electrocompetent *E. coli* containing WT or mutant Cas12a2+3xcrRNA immune system plasmids shown by spot assay. Transformations were plated as 10x serial dilutions between $10^1$ and $10^6$ and the number of colonies in the highest countable spot were used to determine the total number colony forming units (cfu) per 50 ng plasmid.

Transformation fold reduction was calculated as $-\text{Log}_{10}(T_{cfu}/NT_{cfu})$. Active immune systems present a ≥100-fold difference between T and NT transformation efficiency. All transformations were completed to at least two biological replicates. For transformation plates source data, see Supplementary Fig. 1.

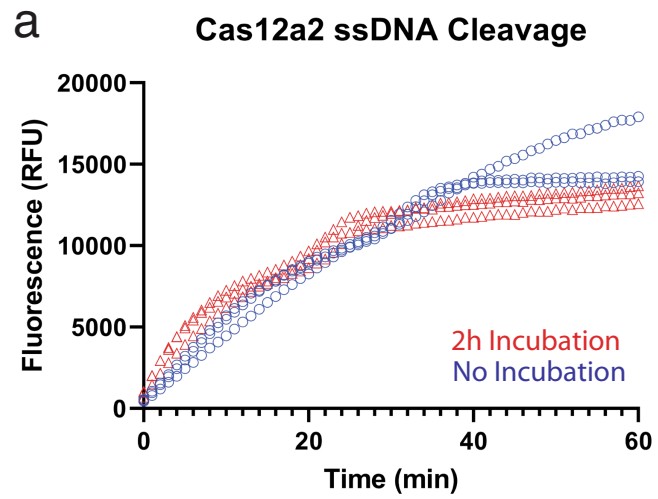

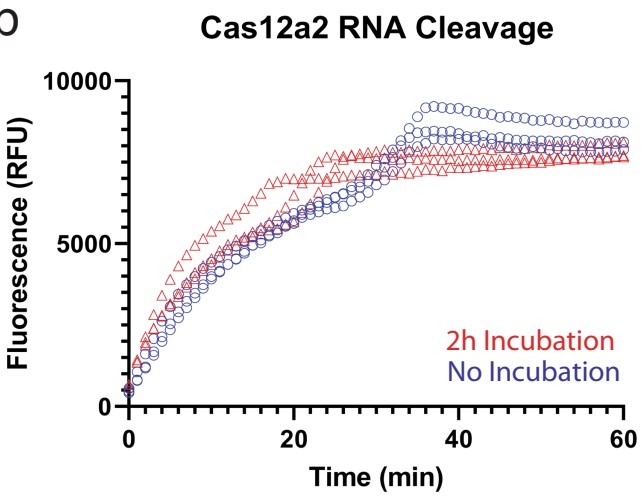

**Extended Data Fig. 10 | Effect of pre-incubating activating RNA with Cas12a2 on collateral ssDNA and ssRNA cleavage.** Cas12a2 cleavage activity after either No or 2 h incubation of Cas12a2:crRNA complex with 2x Target RNA.

**A**. Non-target cleavage of DNAse Alert reporter (IDT) **B**. Non-target cleavage of RNase Alert reporter (IDT). Each experiment was performed in triplicate. For gel source data, see Supplementary Fig. 1.

**Extended Data Table 1 | Cryo-EM data collection and model validation statistics**

| | Binary complex (PDB 8D49, EMD-27178) | Ternary complex (PDB 8D4B, EMD-27180) | Quaternary complex (PDB 8D4A, EMD-27179) |
|---|---|---|---|
| **Data collection and processing** | | | |
| Voltage (kV) | 200 | 300 | 200 |
| Electron exposure (e–/Å$^2$) | 40 | 80 | 40 |
| Defocus range (μm) | | -1.5 to -2.5 | |
| Pixel size (Å) | 0.94 | 0.81 | 0.94 |
| Symmetry imposed | | | |
| Initial particle images (no.) | 981,772 | 2,996,634 | 1,692,836 |
| Final particle images (no.) | 80,528 | 192,639 | 94,805 |
| Map resolution (Å) | 3.20 | 2.92 | 2.74 |
| FSC threshold | 0.143 | 0.143 | 0.143 |
| | <2.5 - >4.5 | <2.5 - >4.5 | <2.5 - >4.5 |
| **Refinement** | | | |
| Initial model used (PDB code) | N/A | N/A | N/A |
| Model resolution (Å) | 3.3 | 3.0 | 2.8 |
| FSC threshold | 0.5 | 0.5 | 0.5 |
| Map sharpening $B$ factor (Å$^2$) | 102.6 | 94.4 | 76.3 |
| Model composition | | | |
| Non-hydrogen atoms | 9264 | 11007 | 11792 |
| Protein residues | 1071 | 1166 | 1207 |
| Nucleotides | 26 | 69 | 91 |
| Ligands | | | Zn: 1. Mg: 2. |
| Water | | | 1 |
| Mean $B$ factors (Å$^2$) | | | |
| Protein | 55.59 | 47.81 | 22.98 |
| Nucleotides | 48.39 | 35.66 | 28.01 |
| Ligand | | | 31.42 |
| Water | | | 12.02 |
| R.m.s. deviations | | | |
| Bond lengths (Å) | 0.009 | 0.005 | 0.006 |
| Bond angles (°) | 0.745 | 0.65 | 0.702 |
| Validation | | | |
| MolProbity score | 1.62 | 1.3 | 1.12 |
| Clashscore | 6.14 | 4.99 | 2.92 |
| Poor rotamers (%) | 0 | 0 | 0 |
| Ramachandran plot | | | |
| Favored (%) | 95.94 | 97.84 | 97.84 |
| Allowed (%) | 4.06 | 2.16 | 2.16 |
| Disallowed (%) | 0 | 0 | 0 |
| Map CC (mask) | 0.80 | 0.83 | 0.86 |

# Reporting Summary

## Statistics

For all statistical analyses, confirm that the following items are present in the figure legend, table legend, main text, or Methods section.

| n/a | Confirmed | |
|---|---|---|
| ☐ | ☒ | The exact sample size (*n*) for each experimental group/condition, given as a discrete number and unit of measurement |
| ☐ | ☒ | A statement on whether measurements were taken from distinct samples or whether the same sample was measured repeatedly |
| ☐ | ☒ | The statistical test(s) used AND whether they are one- or two-sided *Only common tests should be described solely by name; describe more complex techniques in the Methods section.* |
| ☒ | ☐ | A description of all covariates tested |
| ☒ | ☐ | A description of any assumptions or corrections, such as tests of normality and adjustment for multiple comparisons |
| ☐ | ☒ | A full description of the statistical parameters including central tendency (e.g. means) or other basic estimates (e.g. regression coefficient) AND variation (e.g. standard deviation) or associated estimates of uncertainty (e.g. confidence intervals) |
| ☐ | ☒ | For null hypothesis testing, the test statistic (e.g. *F*, *t*, *r*) with confidence intervals, effect sizes, degrees of freedom and *P* value noted *Give P values as exact values whenever suitable.* |
| ☒ | ☐ | For Bayesian analysis, information on the choice of priors and Markov chain Monte Carlo settings |
| ☒ | ☐ | For hierarchical and complex designs, identification of the appropriate level for tests and full reporting of outcomes |
| ☒ | ☐ | Estimates of effect sizes (e.g. Cohen's *d*, Pearson's *r*), indicating how they were calculated |

*Our web collection on statistics for biologists contains articles on many of the points above.*

## Software and code

Policy information about availability of computer code

| Data collection | Data were collected on a FEI Glacios cryo-TEM equipped with a Falcon 4 detector. Data was collected in SerialEM v3.8, with a pixel size of 0.94 Å, a defocus range of -1.5 - -2.5 μm, and a total exposure time of 15s resulting in a total accumulated dose of 40 e/Å2 which was split into 60 EER fractions. |
|---|---|
| Data analysis | Motion correction, CTF estimation and particle picking was performed on-the-fly using cryoSPARC Live v4.0.0-privatebeta.2 and cryoSPARC v3.2. All subsequent data processing was performed in cryoSPARC v3.2. Models were built using COOT v1.0 and relaxed using flexible molecular dynamics fitting with ISOLDE v1.4, then finally subjected to real-space refinement as implemented in PHENIX v1.19.2. All structural figures and movies were generated using ChimeraX v1.0 , apart from the modevectors, which were generated in PyMol v2.5. |

For manuscripts utilizing custom algorithms or software that are central to the research but not yet described in published literature, software must be made available to editors and reviewers. We strongly encourage code deposition in a community repository (e.g. GitHub). See the Nature Portfolio guidelines for submitting code & software for further information.

## Data

Policy information about availability of data

All manuscripts must include a data availability statement. This statement should provide the following information, where applicable:

- Accession codes, unique identifiers, or web links for publicly available datasets
- A description of any restrictions on data availability
- For clinical datasets or third party data, please ensure that the statement adheres to our policy

The atomic models of Cas12a2 binary, tertiary and quaternary complexes have been deposited into the Protein Data Bank (PDB) with codes PDB 8D49, PDB 8D48 and PDB 8D4A, and the corresponding maps have been deposited into the Electron Microscopy Data Bank (EMDB) with codes EMD-29178, EMD-27180 and EMD-27179, respectively. Requests for materials can be sent to R.N.J. (ryan.jackson@usu.edu) and D.W.T. (dtaylor@utexas.edu).

## Human research participants

Policy information about studies involving human research participants and Sex and Gender in Research.

| | |
|---|---|
| Reporting on sex and gender | N/A |
| Population characteristics | N/A |
| Recruitment | N/A |
| Ethics oversight | N/A |

Note that full information on the approval of the study protocol must also be provided in the manuscript.

# Field-specific reporting

Please select the one below that is the best fit for your research. If you are not sure, read the appropriate sections before making your selection.

☒ Life sciences  ☐ Behavioural & social sciences  ☐ Ecological, evolutionary & environmental sciences

For a reference copy of the document with all sections, see nature.com/documents/nr-reporting-summary-flat.pdf

# Life sciences study design

All studies must disclose on these points even when the disclosure is negative.

| | |
|---|---|
| Sample size | A total of ~2000 micrographs were collected for each sample. Each data set contained ~1 million particles. These are typical numbers for cryo-EM data sets to obtain high-resolution structures and provided the required resolution required for our structural analysis and results. |
| Data exclusions | 2D and 3D classification were used to remove damaged particles. This is standard practice in cryo-EM studies and is necessary to obtain homogenous particles for reconstruction. |
| Replication | Cryo-EM datasets were collected on multiple samples in separate imaging sessions with identical overall domain structures. |
| Randomization | No randomization was performed. This study does not require randomization because no animal or human subjects were used. |
| Blinding | No blinding was performed. This study does not require blinding because no animal or human subjects were used. |

# Reporting for specific materials, systems and methods

We require information from authors about some types of materials, experimental systems and methods used in many studies. Here, indicate whether each material, system or method listed is relevant to your study. If you are not sure if a list item applies to your research, read the appropriate section before selecting a response.

## Materials & experimental systems

| n/a | Involved in the study |
|---|---|
| ☒ | ☐ Antibodies |
| ☒ | ☐ Eukaryotic cell lines |
| ☒ | ☐ Palaeontology and archaeology |
| ☒ | ☐ Animals and other organisms |
| ☒ | ☐ Clinical data |
| ☒ | ☐ Dual use research of concern |

## Methods

| n/a | Involved in the study |
|---|---|
| ☒ | ☐ ChIP-seq |
| ☒ | ☐ Flow cytometry |
| ☒ | ☐ MRI-based neuroimaging |

