## [Peer Review File · Nature]

Manuscript Title: RNA targeting unleashes indiscriminate nuclease activity of CRISPR-Cas12a2

Reviewer Comments & Author Rebuttals

Reviewer Reports on the Initial Version:

Referees' comments:

Referee #1:

In this study, Bravo et al. present high-resolution cryo-EM structures of SuCas12a2 in its binary (Cas12a2+crRNA), ternary (Cas12a2+crRNA+RNA target) and quaternary (Cas12a2+crRNA+RNA target+dsDNA collateral substrate) state. SuCas12a2 is an ortholog of type V CRISPR Cas12a2 effectors found in *Sulfolobus solfataricus* sp. and acts as an RNA-guided endonuclease binding complementary target RNAs. Target RNA binding activates SuCas12a2, cleaving ssRNA, ssDNA, and dsDNA substrates in a sequence non-specific fashion, resulting in an abortive infection phenotype (Abi). In this study, the authors reveal the complete activation pathway of SuCas12a2 (from target RNA binding to collateral activity) and identify critical residues responsible for these catalytic activities.

Of interest, the authors identified a unique 'aromatic clamping' mechanism that explains how Cas12a2 can act on DNA duplexes, by bending and local melting of the DNA. They generated mutants of the participating residues and showed that collateral duplex DNA degradation was abolished (while retaining ssRNase and ssDNase nuclease activities). In vivo, this mutant resulted in lower transformation fold reductions, indicative that duplex DNA degradation is one of the main contributors to the Abi phenotype. Finally, they propose that the mutant variant Y1069A can degrade ssDNA, but not ssRNA, or dsDNA could be adapted as an RNA sensing diagnostics tool.

This work presents an interesting twist on the modus operandi of Cas12 effector nucleases, where ssRNA targeting by SuCas12a2 elicits broad and indiscriminate DNase activity. The characterization of the structures presented allows for a deeper insight into the diversity of type V systems and the mechanisms that seem to be geared towards conferring population-level immunity. Access to the structure of SuCas12a2 allows the rational design of novel variants that can be adapted to degrade a spectrum of different substrates, providing opportunities to develop ssRNA targeting diagnostic tools. The results presented are in agreement with the discussions and conclusions made in the manuscript. The experiments are well conducted and described, making this an exciting story that adds to the growing list of evidence that many CRISPR-Cas systems confer population-wide immunity.

Major comment

Since the authors found that "the hybridized region of the target RNA remains intact", do the authors have any clues how long Cas12a2 will stay in its active (collateral damage-inducing) conformation? By referring to it as a "nuclear launch button" (as a side note: this metaphor is, in this reviewer's opinion, not appropriate. How about a "self-destruct button"?) the authors seem to favor the idea that a single target RNA will cause a point of no return, which sounds like a (too?) risky strategy given the flexible targeting requirements of Cas12a2 (as detailed in the Dmytrenko et al. study). In other words, do the authors have evidence that this is indeed the case (i.e. the irreversible activation of Cas12a2)?

Minor comments

Abstract

"...nuclease that performs RNA-guided degradation of non-specific single-stranded (ss)RNA,"
Please consider changing this to "...nuclease that performs RNA-guided, sequence non-specific degradation of single-stranded (ss)RNA,".

Introduction

"...abortive infection (Abi) – that is, dormancy in response to the presence of an invader – to achieve population-level immunity"

Please note that abortive infection is defined as any mechanism by the infected host that prevents the production of viral progeny, which is not restricted to dormancy but also cell death. Please adjust accordingly.

"...Cas12a2 often co-occurs with Cas12a"

The manuscript of Dmytrenko et al., 2022 mentions that only "some" co-occur.

"Cas12a and Cas12a2 sequences bear little resemblance to one another (~10-20%)."

Please specify whether the authors are referring to sequence identity or similarity here.

"...we performed biochemical, structural, and in vivo analyses"

Please put "in vivo" in italics.

Results

"Cas12a2 is insensitive to single mismatches within the entirety of the crRNA but has reduced in vivo activity when truncated on 3' end."

There is no reference on which these claims are based. Perhaps the authors wanted to refer to Dmytrenko et al., 2022 here?

"While other Cas12 proteins undergo conformational changes upon crRNA hybridization (up to ~25Å in for Cas12a 20 and Cas12j 21, but more typically up to ~10 Å²²). The conformational..."
Change period to comma.

"This is in stark contrast to the highly exposed Cas12 RuvC active site, providing a structural mechanism for dsDNA cleavage in trans."

Please specify the Cas12 variant.

Fig. 4: The word "PFS" is partially blocked.

Ext. Data Fig. 4b: The numbering of the PFS is hard to read because of the color.

Referee #2:

The authors report cryo-EM structures of Cas12a2-crRNA binary, Cas12a2-crRNA-ssRNA ternary, and Cas12a2-crRNA-ssRNA-dsDNA quaternary complexes, revealing mechanisms underlying target RNA-activated dsDNA cleavage by Cas12a2. The quaternary Cas12a2-crRNA-ssRNA-dsDNA complex structure provides a structural basis for the non-specific dsDNA degradation. The authors have also determined two pairs of 'aromatic clamp' residues crucial for nuclease activity and substrate selection. This paper presents a set of new data that will be of interest to the field. However, there are deficiencies that must be addressed.

Major points:

1) The labels for domains, residues of the structures in many figures are missing (Fig. 1c, Fig. 2b,c,f, Fig. 3a,c, Fig. 4 and so on); it's difficult to follow what the authors want to present. The

authors should carefully go through the paper and label all the figures properly.

2) Line 91, "The differences in the structural organization of the REC lobe likely allow Cas12a2 to escape targeting by many anti-CRISPR (Acr) proteins that can efficiently shut down Cas12a (Extended Data Fig. 3).": Have the authors tested the inhibition of Cas12a2 nuclease activities by these anti-CRISPR (Acr) proteins in vitro?

3) Line 111, "Cas12a2, crRNA, and a target ssRNA containing a non-self protospacer-flanking sequence (PFS, 5'-GAAAG-3') (Fig. 2)": Could the authors explain why Cas12a2 could be activated by non-self protospacer-flanking sequence but not self sequence, and prefer AAA sequence and tolerate lots of PFS mutants?

4) Line 116, "while the 3' PFS end of the target RNA is gripped by the PI domain, which has now become ordered": The domain color in different should be consistent in the paper. the PI domain can't be found in Fig. 2b. Structure-based mutation of the residues interacting with 5 nt of the PFS may be necessary to figure out the impact on the nuclease activities.

5) Line 120, "allowing Cas12a2 to distinguish self (i.e., complementary to the crRNA 5' handle) from non-self target RNA based on the PFS.": Could the authors explain the mechanism for self and non-self discrimination? Have the authors tried to determine the structure of Cas12a2-crRNA bound to self target RNA?

6) Line 202, "the NCS clamp mutations Y465A and Y1080A reduce and abrogate duplex cleavage, respectively": The cleavage activity is not abrogated, given that the cleavage bands could be still observed.

7) The description and labels for Fig. 2g are not clear; could the authors explain why RNA degradation is predominantly in trans? It seems Cas12a2 can cleave the target both in cis and trans.

8) Line 218, "These data also suggest that duplex degradation is the driving force behind Cas12a2-mediated immunity, as mutants that retained ssRNase and ssDNase activities were not sufficient to provide immunity.": Please explain why the Y465A mutant shows dsDNA cleavage activity (Fig. 3h), but fails to clear the target plasmid in vivo. The data in Fig. 3i is not convincing.

Minor points:

1) It would be better all the structures in Ext. Data Fig. 2 are shown in the same orientation. The domains should be labeled in the structures.

2) Line 100, "In contrast, Cas12a2 is insensitive to single mismatches within the entirety of the crRNA but has reduced in vivo activity when truncated on 3' end.": Please add the reference.

3) Please label the residues and the interactions in Ext. Data Fig. 4, such as the hydrogen bonds could be labeled by dashed lines.

4) Line 126, "In contrast, Cas12a2 is exclusively activated upon recognition of an appropriate PFS sequence. Cas12a2 is unable to degrade nucleic acids in the absence of a suitable target RNA (Extended Data Fig. 4).": Please specify the words "appropriate" "suitable". Have the authors tested PFS mutants in vitro?

5) Please label the purple element in Ext. Data Fig. 5b.

6) Line 139, "Inspection of the RuvC active site in the autoinhibited binary complex reveals that the catalytic triad (D848, E1063, D1213) is buried within a solvent-excluded pocket.": Please add

the reference.

7) Line 146, "The conformational rearrangements we observe for Cas12a2 are considerably larger, highlighting the distinct activation mechanism of Cas12a2": Please specify the scale of the conformational changes.

8) Line 160, "The lack of a Nuc domain and the presence of a highly exposed RuvC active site in the ternary structure thus explain why Cas12a2 collateral nuclease activation results in an Abi phenotype (Dmytrenko 2022) while Cas12a collateral ssDNase activity does not play a role in bacterial immunity": Could the lack of a Nuc domain and the presence of a highly exposed RuvC active site in the ternary structure explain.....?

9) Please label residues and nucleotides in Fig. 3. It's difficult to see the labels in Fig. 3h.

10) Line 179, "we determined a 2.7 Å-resolution structure of crRNA-guided Cas12a2 bound to both an activating target RNA and a collateral dsDNA substrate analog (Fig. 3)": Please specify the analog.

11) Fig. 4 only shows the target is cleaved in trans; however, the complex also cleaves RNA target both in cis and in trans.

12) Ext. Data Fig. 8 wasn't cited at all.

Author Rebuttals to Initial Comments:

Response to Editor (appropriate to share with referees):

Thank you for handling the review of the manuscript. We have addressed all of the comments of the referees below. The feedback was overwhelmingly positive in general, and the comments were very constructive. We believe that the changes and incorporations from their feedback has significantly strengthened the manuscript.

Referee #1:

In this study, Bravo et al. present high-resolution cryo-EM structures of SuCas12a2 in its binary (Cas12a2+crRNA), ternary (Cas12a2+crRNA+RNA target) and quaternary (Cas12a2+crRNA+RNA target+dsDNA collateral substrate) state. SuCas12a2 is an ortholog of type V CRISPR Cas12a2 effectors found in *Sulfolobus solfataricus* sp. and acts as an RNA-guided endonuclease binding complementary target RNAs. Target RNA binding activates SuCas12a2, cleaving ssRNA, ssDNA, and dsDNA substrates in a sequence non-specific fashion, resulting in an abortive infection phenotype (Abi). In this study, the authors reveal the complete activation pathway of SuCas12a2 (from target RNA binding to collateral activity) and identify critical residues responsible for these catalytic activities.

Of interest, the authors identified a unique 'aromatic clamping' mechanism that explains how Cas12a2 can act on DNA duplexes, by bending and local melting of the DNA. They generated mutants of the participating residues and showed that collateral duplex DNA degradation was abolished (while retaining ssRNase and ssDNase nuclease activities). In vivo, this mutant resulted in lower transformation fold reductions, indicative that duplex DNA degradation is one of the main contributors to the Abi phenotype. Finally, they propose that the mutant variant Y1069A can degrade ssDNA, but not ssRNA, or dsDNA could be adapted as an RNA sensing diagnostics tool.

This work presents an interesting twist on the modus operandi of Cas12 effector nucleases, where ssRNA targeting by SuCas12a2 elicits broad and indiscriminate DNase activity. The characterization of the structures presented allows for a deeper insight into the diversity of type V systems and the mechanisms that seem to be geared towards conferring population-level immunity. Access to the structure of SuCas12a2 allows the rational design of novel variants that can be adapted to degrade a spectrum of different substrates, providing opportunities to develop ssRNA targeting diagnostic tools. The results presented are in agreement with the discussions and conclusions made in the manuscript. The experiments are well conducted and described, making this an exciting story that adds to the growing list of evidence that many CRISPR-Cas systems confer population-wide immunity.

Major comment

Since the authors found that "the hybridized region of the target RNA remains intact", do the authors have any clues how long Cas12a2 will stay in its active (collateral damage-inducing)

conformation? By referring to it as a “nuclear launch button” (as a side note: this metaphor is, in this reviewer’s opinion, not appropriate. How about a “self-destruct button”?) the authors seem to favor the idea that a single target RNA will cause a point of no return, which sounds like a (too?) risky strategy given the flexible targeting requirements of Cas12a2 (as detailed in the Dmytrenko et al. study). In other words, do the authors have evidence that this is indeed the case (i.e. the irreversible activation of Cas12a2)?

We appreciate the reviewer’s comment, and we agree that the language is indeed inappropriate after further reflection. We have changed the language in our manuscript to “self-destruct button”.

We based this discussion point on the hypothesis that the cognate target strand RNA does not dissociate from Cas12a2 (our ternary and quaternary structures were determined 30 and 60 minutes after addition of activating TS RNA and no dissociated complexes were observed). However, in cells, there are likely other mechanisms that may mitigate Cas12a2 activation over time, such as protein turnover.

Additionally, Williams et al, 2022 demonstrated that in the case of Cas13-mediated cellular dormancy, once the phage DNA encoding the Cas13-targeted transcripts are degraded by cellular restriction-modification systems, the cell population becomes resurrected, indicating that Cas13 at some point will become inactive. These references have been added to the manuscript.

To directly address the question of whether Cas12a2 remains active long after target RNA binding, we have included new experimental data to our manuscript (as Extended Data Fig. 10). In our revised manuscript, we test this possibility by activating Cas12a2 with target RNA and measuring ssDNA and ssRNA cleavage *in trans* under different pre-incubation conditions (no pre-incubation and 2h pre-incubation). Pre-incubation of Cas12a2 with activating target RNA does not result in discernable differences in collateral cleavage, indicating that Cas12a2 can remain active for hours after binding of a suitable target RNA.

Extended Data Fig. 10 | Effect of pre-incubating activating RNA with Cas12a2 on collateral ssDNA and ssRNA cleavage. Cas12a2 cleavage activity after either No or 2 H incubation of Cas12a2:crRNA complex with 2x Target RNA. **A.** Non-target cleavage of DNase Alert reporter (IDT) **B.** Non-target cleavage of RNase Alert reporter (IDT).

However, since we cannot comment on Cas12a2 turnover *in vivo*, we have toned down our language. We now add: “The consequences of Cas12a2 activation may be mitigated through protein

degradation and turnover coupled with the removal of phage transcript-encoding DNA by other defense systems (e.g. Restriction-Modification systems and DNA-targeting CRISPR-Cas systems)”

Minor comments

Abstract

“...nuclease that performs RNA-guided degradation of non-specific single-stranded (ss)RNA,”
Please consider changing this to “...nuclease that performs RNA-guided, sequence non-specific degradation of single-stranded (ss)RNA,”.

This has been changed.

Introduction

“...abortive infection (Abi) – that is, dormancy in response to the presence of an invader – to achieve population-level immunity”

Please note that abortive infection is defined as any mechanism by the infected host that prevents the production of viral progeny, which is not restricted to dormancy but also cell death. Please adjust accordingly.

This has been changed.

“...Cas12a2 often co-occurs with Cas12a”

The manuscript of Dmytrenko et al., 2022 mentions that only “some” co-occur.

This has been changed.

“Cas12a and Cas12a2 sequences bear little resemblance to one another (~10-20%).”

Please specify whether the authors are referring to sequence identity or similarity here.

This is sequence identity.

“...we performed biochemical, structural, and in vivo analyses”

Please put “in vivo” in italics.

This has been changed.

Results

“Cas12a2 is insensitive to single mismatches within the entirety of the crRNA but has reduced in vivo activity when truncated on 3’ end.”

There is no reference on which these claims are based. Perhaps the authors wanted to refer to Dmytrenko et al., 2022 here?

This has been added.

“While other Cas12 proteins undergo conformational changes upon crRNA hybridization (up to ~25Å in for Cas12a 20 and Cas12j 21, but more typically up to ~10 Å²²). The conformational...”

Change period to comma.

This has been changed.

“This is in stark contrast to the highly exposed Cas12 RuvC active site, providing a structural mechanism for dsDNA cleavage in trans.”

Please specify the Cas12 variant.

This has been added.

Fig. 4: The word “PFS” is partially blocked.

This has been fixed.

Ext. Data Fig. 4b: The numbering of the PFS is hard to read because of the color.

This has been changed.

Referee #2:

The authors report cryo-EM structures of Cas12a2-crRNA binary, Cas12a2-crRNA-ssRNA ternary, and Cas12a2-crRNA-ssRNA-dsDNA quaternary complexes, revealing mechanisms underlying target RNA-activated dsDNA cleavage by Cas12a2. The quaternary Cas12a2-crRNA-ssRNA-dsDNA complex structure provides a structural basis for the non-specific dsDNA degradation. The authors have also determined two pairs of ‘aromatic clamp’ residues crucial for nuclease activity and substrate selection. This paper presents a set of new data that will be of interest to the field. However, there are deficiencies that must be addressed.

Major points:

1) The labels for domains, residues of the structures in many figures are missing (Fig. 1c, Fig. 2b,c,f, Fig. 3a,c, Fig. 4 and so on); it’s difficult to follow what the authors want to present. The authors should carefully go through the paper and label all the figures properly.

This has been added to most panels. Since Fig. 3a is very similar to Fig. 2b (which now has labels), aside from the *trans* dsDNA (which is labeled), we have not annotated the domains, as this makes the panel difficult to interpret. Fig. 3c is labeled.

2) Line 91, “The differences in the structural organization of the REC lobe likely allow Cas12a2 to escape targeting by many anti-CRISPR (Acr) proteins that can efficiently shut down Cas12a (Extended Data Fig. 3).”: Have the authors tested the inhibition of Cas12a2 nuclease activities by these anti-CRISPR (Acr) proteins in vitro?

This has been demonstrated *in vivo*, in the accompanying manuscript. This reference has now been added to support this statement.

3) Line 111, “Cas12a2, crRNA, and a target ssRNA containing a non-self protospacer-flanking sequence (PFS, 5’-GAAAG-3’) (Fig. 2)”: Could the authors explain why Cas12a2 could be activated by non-self

protospacer-flanking sequence but not self sequence, and prefer AAA sequence and tolerate lots of PFS mutants?

This is an excellent suggestion. We attempted to determine a structure of Cas12a2 bound to a self PFS-containing target strand (TS) RNA. Unfortunately, in our hands, we found that the complex aggregated and became insoluble upon mixing of the Cas12a2 binary complex with this TS RNA. Since the self PFS is complementary to the 5' handle of the crRNA, we speculate that the addition of self-PFS TS may partially pair with the crRNA, thus disrupting contacts with the Cas12a2 WED domain, rendering the complex unstable.

The precise mechanism of self PFS-mediated Cas12a2 autoinhibition is fascinating and will be the subject of future studies.

4) Line 116, "while the 3' PFS end of the target RNA is gripped by the PI domain, which has now become ordered": The domain color in different should be consistent in the paper. the PI domain can't be found in Fig. 2b. Structure-based mutation of the residues interacting with 5 nt of the PFS may be necessary to figure out the impact on the nuclease activities.

We believe the domain color scheme is consistent throughout the manuscript. The PI domain is light pink. In Fig. 1b, the cryo-EM structure is shown colored by protein domain from the atomic model as shown in Fig. 1c. Since the density for the PI domain in the binary complex is too weak to accurately model, this region has been depicted in grey. In all subsequent figures where the PI is ordered it is colored pink. Labels have now been added in Fig. 2b.

Our manuscript already includes data that demonstrates that removal of the PI domain does not disrupt the overall structure of Cas12a2 (Extended Data Fig. 4f), yet prevents activation *in vivo* (Fig. 3i). Furthermore, we show that removal of the PFS prevents target cleavage (Extended Data Fig. 4d). The referee is pointing toward fascinating studies of PI point mutants that may have altered PFS sensing activity. These studies would be the basis for a future manuscript, and are beyond the scope of this manuscript.

5) Line 120, "allowing Cas12a2 to distinguish self (i.e., complementary to the crRNA 5' handle) from non-self target RNA based on the PFS.": Could the authors explain the mechanism for self and non-self discrimination? Have the authors tried to determine the structure of Cas12a2-crRNA bound to self target RNA?

This has been addressed above, in response to referee comment 3.

6) Line 202, "the NCS clamp mutations Y465A and Y1080A reduce and abrogate duplex cleavage, respectively": The cleavage activity is not abrogated, given that the cleavage bands could be still observed.

This has been adjusted in the text. We also have additional data (see response to point 8) that shows that both of the NCS clamp mutants fail to fully degrade supercoiled plasmid DNA, but are still able to nick and linearize the substrate, albeit slower than wild-type Cas12a2.

7) The description and labels for Fig. 2g are not clear; could the authors explain why RNA degradation is predominantly in trans? It seems Cas12a2 can cleave the target both in cis and trans.

Our claim that *trans* cleavage is likely more often than *cis* is based on our structural data, and an accompanying protection assay that demonstrates that the bound RNA target is protected from degradation when bound to the gRNA. The ends of the bound RNA are trimmed. Admittedly, some trimming could be performed by ends of RNA target falling into the active site of the exact same Cas12a2 nuclease that was active. But steric strain and distances between the Cas12a2 active sites do not allow for the complete trimming of the end that we observe. Thus, even if some bound RNAs are initially cleaved in *cis*, the final trimming appears to be mediated in *trans*.

We have changed the language in the manuscript to more accurately reflect our reasoning.

8) Line 218, “These data also suggest that duplex degradation is the driving force behind Cas12a2-mediated immunity, as mutants that retained ssRNase and ssDNase activities were not sufficient to provide immunity.”: Please explain why the Y465A mutant shows dsDNA cleavage activity (Fig. 3h), but fails to clear the target plasmid *in vivo*. The data in Fig. 3i is not convincing.

We have now measured supercoiled plasmid cleavage by Cas12a2 and the four aromatic clamp mutants. All four mutants – including Y465A – have reduced plasmid cleavage *in vitro*. Consistent with our *in vitro* cleavage data, the Y465A still supports plasmid nicking and/or linearization but is unable to fully degrade supercoiled plasmid in a 1-hr incubation. Notably, the small, linear, FAM-labeled dsDNA substrate used in Fig. 3 would be amenable to nicking activities. The impaired cleavage of supercoiled plasmid DNA provides a rationale to explain why the Y465A mutant has a clear immune deficiency *in vivo* – by preventing supercoiled plasmid degradation, Cas12a2 immunity is reduced.

We believe that these data strengthen our manuscript, and they are now included in the Extended Data (Extended Data Fig. 8).

a.

Supercoiled Plasmid Cleavage

b.

Extended Data Fig. 8 | Effect of clamp residue mutations on supercoiled DNA cleavage *in vitro*.

Rate of supercoiled DNA cleavage by wild-type Cas12a2 and the four aromatic clamp mutants described in Fig. 3. WT Cas12a2 cleaves supercoiled plasmid at a rate of 0.7 min^{-1} , while all four mutants reduce cleavage. Y465A retains an ability (albeit much slower than WT Cas12a2) to nick and/or linearize plasmid, but fails to fully degrade plasmid DNA. **a**, Quantification of fraction of supercoiled pUC19 cleaved over time. **B**, Representative agarose gels of supercoiled pUC19 cleavage by WT Cas12a2 or aromatic clamp mutants over time.

We have also added the following to the results section:

“We further tested the effects of these point mutations in Cas12a2 on supercoiled plasmid DNA cleavage activity and found that all had severely reduced activity (**Extended Data Fig. 8**). Interestingly, the Y465A NCS mutant is able to nick and/or linearize plasmid, but is unable to further degrade DNA to an extent similar to WT, providing a rationale for the *in vivo* effects of this mutant.”

Minor points:

1) It would be better all the structures in Ext. Data Fig. 2 are shown in the same orientation. The domains should be labeled in the structures.

If these structures were in the same orientation, it would not be possible to see both the trajectory of the crRNA:TS duplex and the crRNA 5' handle. We have added a schematic to Extended Data Fig. 2g.

2) Line 100, “In contrast, Cas12a2 is insensitive to single mismatches within the entirety of the crRNA but has reduced *in vivo* activity when truncated on 3' end.”: Please add the reference.

This has been added.

3) Please label the residues and the interactions in Ext. Data Fig. 4, such as the hydrogen bonds could be labeled by dashed lines.

This has been added.

4) Line 126, “In contrast, Cas12a2 is exclusively activated upon recognition of an appropriate PFS sequence. Cas12a2 is unable to degrade nucleic acids in the absence of a suitable target RNA (Extended Data Fig. 4).”: Please specify the words “appropriate” “suitable”. Have the authors tested PFS mutants *in vitro*?

This is in the accompanying manuscript. The PFS motif was identified *in vivo*, where a library of plasmids with all 1024 possible five-nucleotide PFS sequences were transformed into cells expressing Cas12a2 and a targeting gRNA, resulting in depletion of targets with recognized PFS sequences. Individual PFS trends were then confirmed and validated by testing individual PFS sequences *in vivo*. A reference to the accompanying paper has now been added to indicate this.

We feel that an *in vitro* confirmation of the various identified PFS sequences is beyond the scope of this paper, although we agree with the line of questioning the reviewer is making. We plan to investigate the role of the PFS sequence on Cas12a2 activation in future studies.

5) Please label the purple element in Ext. Data Fig. 5b.

This has been added.

6) Line 139, “Inspection of the RuvC active site in the autoinhibited binary complex reveals that the catalytic triad (D848, E1063, D1213) is buried within a solvent-excluded pocket.”: Please add the reference.

This has been added.

7) Line 146, “The conformational rearrangements we observe for Cas12a2 are considerably larger, highlighting the distinct activation mechanism of Cas12a2”: Please specify the scale of the conformational changes.

This has been added.

8) Line 160, “The lack of a Nuc domain and the presence of a highly exposed RuvC active site in the ternary structure thus explain why Cas12a2 collateral nuclease activation results in an Abi phenotype (Dmytrenko 2022) while Cas12a collateral ssDNase activity does not play a role in bacterial immunity”: Could the lack of a Nuc domain and the presence of a highly exposed RuvC active site in the ternary structure explain.....?

We have added the following:

“The Nuc domain may act as a physical barrier to the RuvC active site in other Cas12 enzymes (including Cas12a), limiting cleavage *in trans*.”

9) Please label residues and nucleotides in Fig. 3. It’s difficult to see the labels in Fig. 3h.

The residues are labeled. Since the DNA substrate is bound in multiple registers due to lack of sequence-specific binding, we cannot annotate the nucleotides.

The labels in Fig 3h have been resized accordingly.

10) Line 179, “we determined a 2.7 Å-resolution structure of crRNA-guided Cas12a2 bound to both an activating target RNA and a collateral dsDNA substrate analog (Fig. 3).”: Please specify the analog.

This has been added.

11) Fig. 4 only shows the target is cleaved in trans; however, the complex also cleaves RNA target both in cis and in trans.

This has been added.

12) Ext. Data Fig. 8 wasn’t cited at all.

This has been added.

Reviewer Reports on the First Revision:

Referees' comments:

Referee #1:

The authors thoroughly addressed my and the other reviewer's questions and added additional data to substantiate their conclusions even further.

A few last remaining issues:

1. The authors should define the error bars in Fig. 3i, Ext. Data Fig. 8a and mention how many replicates were used for these figures.
2. Throughout the whole manuscript change "H" into "h" (lower case) when referring to hours.
3. "Protein was preheated at 37C" , add degree symbol after "37".
4. Revise "Time points were removed at..." (e.g. "Samples were taken at time points").

Referee #2:

The authors have addressed all my concerns with additional experiments and revised descriptions.

Author Rebuttals to First Revision:

Referee #1:

The authors thoroughly addressed my and the other reviewer's questions and added additional data to substantiate their conclusions even further.

We thank the review for their helpful feedback. Our manuscript has been greatly strengthened by their comments.

A few last remaining issues:

1. The authors should define the error bars in Fig. 3i, Ext. Data Fig. 8a and mention how many replicates were used for these figures.

This has been added.

2. Throughout the whole manuscript change "H" into "h" (lower case) when referring to hours.

This has been fixed.

3. "Protein was preheated at 37C" , add degree symbol after "37".

This has been fixed.

4. Revise "Time points were removed at..." (e.g. "Samples were taken at time points").

This has been fixed.

Referee #2:

The authors have addressed all my concerns with additional experiments and revised descriptions.

We thank the review for their helpful feedback. Our manuscript has been greatly strengthened by their comments.